# A survey of methane point source emissions from coal mines in Shanxi province of China using AHSI on board Gaofen-5B

**Zhonghua He**[1]**, Ling Gao**[2]**, Miao Liang**[3]**, and Zhao-Cheng Zeng**[4]

[1]Zhejiang Climate Centre, Zhejiang Meteorological Bureau, Hangzhou, 310052, China
[2]National Satellite Meteorological Centre, China Meteorological Administration, Beijing, 100081, China
[3]Meteorological Observation Centre, China Meteorological Administration, Beijing, 100081, China
[4]Institute of Remote Sensing and GIS, School of Earth and Space Sciences, Peking University, Beijing, 100871, China

**Correspondence:** Zhao-Cheng Zeng (zczeng@pku.edu.cn)

**Abstract.** Satellite-based detection of methane ($CH_4$) point sources is crucial in identifying and mitigating anthropogenic emissions of $CH_4$, a potent greenhouse gas. Previous studies have indicated the presence of $CH_4$ point source emissions from coal mines in Shanxi, China, which is an important source region with large $CH_4$ emissions, but a comprehensive survey has remained elusive. This study aims to conduct a survey of $CH_4$ point sources over Shanxi's coal mines based on observations of the Advanced Hyperspectral Imager (AHSI) on board the Gaofen-5B satellite (GF-5B/AHSI) between 2021 and 2023. The spectral shift in centre wavelength and change in full width at half-maximum (FWHM) from the nominal design values are estimated for all spectral channels, which are used as inputs for retrieving the enhancement of the column-averaged dry-air mole fraction of $CH_4$ ($\Delta XCH_4$) using a matched-filter-based algorithm. Our results show that the spectral calibration on GF-5B/AHSI reduced estimation biases of the emission flux rate by up to 5.0 %. We applied the flood-fill algorithm to automatically extract emission plumes from $\Delta XCH_4$ maps. We adopted the integrated mass enhancement (IME) model to estimate the emission flux rate values from each $CH_4$ point source. Consequently, we detected $CH_4$ point sources in 32 coal mines with 93 plume events in Shanxi province. The estimated emission flux rate ranges from $761.78 \pm 185.00$ to $12\,729.12 \pm 4658.13\,\mathrm{kg\,h^{-1}}$. Our results show that wind speed is the dominant source of uncertainty contributing about 84.84 % to the total uncertainty in emission flux rate estimation. Interestingly, we found a number of false positive detections due to solar panels that are widely spread in Shanxi. This study also evaluates the accuracy of wind fields in ECMWF ERA5 reanalysis by comparing them with a ground-based meteorological station. We found a large discrepancy, especially in wind direction, suggesting that incorporating local meteorological measurements into the study $CH_4$ point source are important to achieve high accuracy. The study demonstrates that GF-5B/AHSI possesses capabilities for monitoring large $CH_4$ point sources over complex surface characteristics in Shanxi.

## 1 Introduction

Due to its potent radiative forcing and relatively short lifespan of about a decade, methane ($CH_4$), the second most significant anthropogenic greenhouse gas after atmospheric carbon dioxide, is an effective target that attracts increasing attention for emission reduction and climate change mitigation (IPCC, 2021). Human-activity-related sources of atmospheric $CH_4$ primarily include agricultural activities like livestock farming and rice cultivation, industrial products and processes such as petroleum, natural gas, and coal extraction, as well as landfills and waste management (Lu et al., 2022). Among these, industrial activities related to fossil fuel production contribute nearly 35 % to global anthropogenic $CH_4$ emissions (Saunois et al., 2020), not only triggering the greenhouse effect but also leading to significant energy wastage (Chen et al., 2023). Methane emissions escaping from energy production activities primarily stem from industrial infrastructure emissions, such as wells, collection and compression stations, storage tanks, pipelines, and process-

ing plants, easily forming "point sources" of $CH_4$ emissions (Varon et al., 2019). With the destruction of geological processes involved in mining activities, the release of coalbed methane captured in coal seams and surrounding rock strata forms the point source of $CH_4$ emission from coal mines (Zheng et al., 2019). These emission plumes of gas release from point sources contain high concentrations of $CH_4$ over relatively small surface areas (Duren et al., 2019). The overall plumes formed by point source emissions exhibit a notable heavy-tailed distribution (Irakulis-Loitxate et al., 2021). However, due to the comprehensive effect of the emission magnitude, land cover types, and wind speed and direction, these plumes often show different characteristics across different time and space changes (Sánchez-García et al., 2022), which makes the plume detection and emission estimation challenging. Given that such emissions contribute significantly to regional $CH_4$ emissions (Frankenberg et al., 2016), it is important to have accurate detection and estimation. As atmospheric $CH_4$ is colourless and odourless, coupled with the strong uncertainty in the temporal and spatial distribution of point source emissions, satellite remote sensing using high-resolution spectroscopy has become a crucial means for detecting $CH_4$ point source emissions due to its high sensitivity, wide coverage, and high revisit capabilities (Pandey et al., 2021).

Satellite observations for detecting global atmospheric $CH_4$ concentrations with high spatiotemporal resolution provide data support for accounting and assessing reduction measures (Jacob et al., 2022). Satellite detection and quantification of $CH_4$ super-emitters were first demonstrated in the 2015 Aliso Canyon blowout incident using the Hyperion Imaging Spectrometer on board EO-1 (Thompson et al., 2016). Satellites with high spatial resolution but with moderate spectral resolution have successfully detected and traced $CH_4$ point source emissions. The current in-orbit satellites include GHGSat operated by a private company in Canada (2016–present; Jervis et al., 2021), Italy's PRISMA (2019–present; Guanter et al., 2021), China's Gaofen-5B (GF-5) and ZY-1 satellites (Irakulis-Loitxate et al., 2021), NASA's EMIT (Thorpe et al., 2023), and the German EnMAP mission (Guanter et al., 2015). While multispectral (Landsat 8/9, Sentinel-2, and WorldView-3) and coarse-resolution, high-spectral satellites (Sentinel-5P TROPOMI) have also been widely validated for detecting extra-large $CH_4$ plumes (Ehret et al., 2022; Varon et al., 2021; Sánchez-Garcia et al., 2022; Lauvaux et al., 2022), limitations in spectral or spatial resolution result in differences in detection sensitivity, estimation uncertainty, and tracing capabilities. The first-generation Advanced Hyperspectral Imager (AHSI) on board China's GF-5A (GF-5A/AHSI) exhibits high capabilities in detecting $CH_4$ point source emissions. As shown in Irakulis-Loitxate et al. (2021), 37 unexpected emission point sources exceeding $500 \, \mathrm{kg\,h^{-1}}$ can be identified in the Permian Basin oil and gas fields using a total of 30 images from GF-5A and PRISMA satellites during several days in 2019 and 2020, il-

lustrating the potential of AHSI in regional $CH_4$ point source survey. To estimate emissions from $CH_4$ point source, these studies typically employ spectral matching filtering method to derive $CH_4$ increment ($XCH_4$) and then estimate flux rate using integrated mass enhancement (IME) model (Varon et al., 2018). These studies have previously provided available techniques in the identification of point sources on local or national scales (e.g. Algeria, Permian, China, and USA) and flux estimation and uncertainty analyses for these point sources (Guanter et al., 2021; Irakulis-Loitxate et al., 2021).

As the world's largest coal producer, China contributes 50.7 % of the global coal production in 2020, making it one of the largest emitters of $CH_4$ from coal mining (Z. Chen et al., 2022), especially in Shanxi province, where most underground coal mines are located (Qin et al., 2023). However, due to the influence of complex surface conditions on the background spectral characteristics, satellite observations exhibit notably lower sensitivity in the detection of $CH_4$ point source emissions in Shanxi compared to other regions with more homogeneous land surfaces (Sánchez-García et al., 2022; Guanter et al., 2021). In addition, the wind fields from reanalysis datasets may be subject to high uncertainty due to the complex terrain in Shanxi, leading to highly uncertain emission flux rate estimation (Jongaramrungruang et al., 2021). Although TROPOMI (TROPOspheric Monitoring Instrument) imagery and convolutional neural networks have been shown to effectively detect potential large $CH_4$ emission point sources globally (Schuit et al., 2023), the specific localisation and tracing of $CH_4$ emission point sources in China remain difficult due to the limitations of coarse spatial resolution and complex regional backgrounds, warranting further surveying efforts.

This study aims to conduct a survey of the $CH_4$ point source plumes in Shanxi by developing a framework to detect and estimate emissions flux rate using the latest hyperspectral observations from GF-5B/AHSI from 2021 to 2023. Specifically, this study focuses on (1) quantifying the impact of the shift in spectral wavelength and the change in spectral instrument line shape (ILS) from the nominal design values for the spectral channels of GF-5/AHSI on $CH_4$ retrieval and emission estimation, (2) identifying $CH_4$ point source plumes using the matched-filter method, (3) automating the segmentation of emission plumes from the retrieved $CH_4$ enhancement maps, (4) estimating emissions flux rate from point sources using IME method, and (5) understanding the spatial and temporal patterns of $CH_4$ emissions from point sources in Shanxi.

## 2    Study area and used datasets

### 2.1    Study area

Shanxi province is the most extensively mined region in China, harbouring nearly half of the nation's suspected point

sources based on TROPOMI observations (Schuit et al., 2023). It stands as a typical area for $CH_4$ point source emissions in China and has been a focal point in prior comparative studies on point source emissions (Sánchez-Garcia et al., 2022; Guanter et al., 2021). Shanxi province (Fig. 1), situated in northern China, experiences a temperate continental monsoon climate characterised by cold, dry winters and hot, humid summers. The region boasts diverse topography, comprising mountains, plateaus, and basins. Consequently, the stable atmospheric conditions during winter can lead to the accumulation of pollutants closer to the ground, impacting the detection of $CH_4$ emissions. Although the region has strict rules in regulating the process of $CH_4$, a byproduct of coal mining, underground coal mines in Shanxi release $CH_4$ to the atmosphere from mine venting. Therefore, the identification of these plumes will help mitigate $CH_4$ emissions over this region.

## 2.2 GF-5B/AHSI dataset

The GF-5B satellite is the second satellite of the Gaofen-5 series and was launched on 7 September 2021. It has accumulated over 2 years' worth of global observational data to date. Equipped with the Advanced Hyperspectral Imager (AHSI), it can capture spectral information spanning 400 to 2500 nm, with a spatial resolution of 30 m over a 60 km swath, encompassing 330 spectral channels with spectral resolutions of 5 and 10 nm in the visible near-infrared (VNIR) and short-wave infrared (SWIR) spectroscopy, respectively (Y.-N. Liu et al., 2019). Its relatively high signal-to-noise ratio (around 500 in the SWIR) presents notable advantages in detecting $CH_4$ point source emissions (Irakulis-Loitxate et al., 2021). The retrieval of the enhancement of column-averaged dry-air mole fraction of $CH_4$ ($\Delta XCH_4$) relies primarily on strong $CH_4$ absorption features near 2300 nm, while the 2100 to 2450 nm spectral window of the GF-5B/AHSI demonstrates higher sensitivity to $XCH_4$ variations, thereby possessing enhanced capabilities for precise $CH_4$ concentration inversion. This study focuses on Shanxi province, using images from 111 GF-5B/AHSI scenes covering suspected point sources from September 2021 to September 2023, with a cloud cover of less than 10 %, employed for $\Delta XCH_4$ inversion and point source identification (Fig. 1b). These images cover the major emission hotspots, as identified by TROPOMI data (Schuit et al., 2023). Noted that, in Shanxi, the overpassing time of GF-5/AHSI is around 11:00–12:00 Beijing time (BJT).

The SWIR imagery in the AHSI band employs a strategic arrangement of four strips. Each SWIR strip corresponds to a 15 km ground swath, resulting in a continuous 60 km swath width across the satellite orbit with four images combined. This configuration yields a total of 2012 px (including 36 overlapped pixels) along the spatial dimension of the SWIR detectors (Y.-N. Liu et al., 2019). Therefore, the target inside the overlapped pixels could be observed twice in 8 s.

## 2.3 Auxiliary data

Methane point source detection and emission estimation involve various auxiliary datasets, mainly including (1) ultra-high-resolution surface imagery for checking false positive in point source detection, (2) wind fields information for estimate emissions from point source plumes, and (3) digital elevation model (DEM) data for the geometric correction of AHSI imagery. High-resolution surface imagery is an indispensable dataset in point source identification and serves as direct evidence for distinguishing interference signals. The high-resolution imagery used in this study primarily comes from Google Earth. Wind speed data are a critical parameter for calculating emission flux rates. The study utilised $U_{10}$ hourly wind speed reanalysis products from ECMWF ERA5, with a spatial resolution of $0.25° \times 0.25°$ (Muñoz-Sabater et al., 2021). Terrain data are crucial for the geometric correction of AHSI imagery, directly impacting the positioning and identification of $\Delta XCH_4$ plume signals. The study used DEM data from SRTM (Farr et al., 2007) with a spatial resolution of 30 m. Additionally, the study obtained hourly meteorology monitoring data, including wind speed and wind direction, from January 2021 to September 2023 from three national meteorological stations in Yangquan, Changzhi, and Jincheng (Fig. 1), obtained from the China Meteorological Administration Data Centre. These data were compared with ERA5's $U_{10}$ hourly wind speed reanalysis products to investigate the uncertainty in the ERA5's wind field.

## 3 Methods

The retrieval of $\Delta XCH_4$ and estimation of emission flux rate from high-resolution hyperspectral data have been implemented in many previous studies (e.g. Cusworth et al., 2020; Guanter et al., 2021; Huang et al., 2020; Varon et al., 2018; Frankenberg et al., 2016) using a combination of matched-filter method and the IME model. This study primarily applies this combined approach to survey the $CH_4$ point source emissions in Shanxi using GF-5B/AHSI. In addition, this study focuses on the quantification the impact of the spectral shift and the change in spectral ILS on the point source emission estimation, the automation of the segmentation of emission plumes from the retrieved $CH_4$ enhancement maps, and the analysis of the spatial and temporal patterns of $CH_4$ emissions from point sources in Shanxi.

### 3.1 $\Delta XCH_4$ retrieval using matched-filter method

#### 3.1.1 Spectral calibration of GF-5B/AHSI

Spectral shift in the centre wavelength and change in full width at half-maximum (FWHM) relative to the nominal spectral calibration for spectral channels significantly affects the retrieval results of $\Delta XCH_4$ using the spectral matched-filter method (e.g. Guanter et al., 2021). The spectral shift

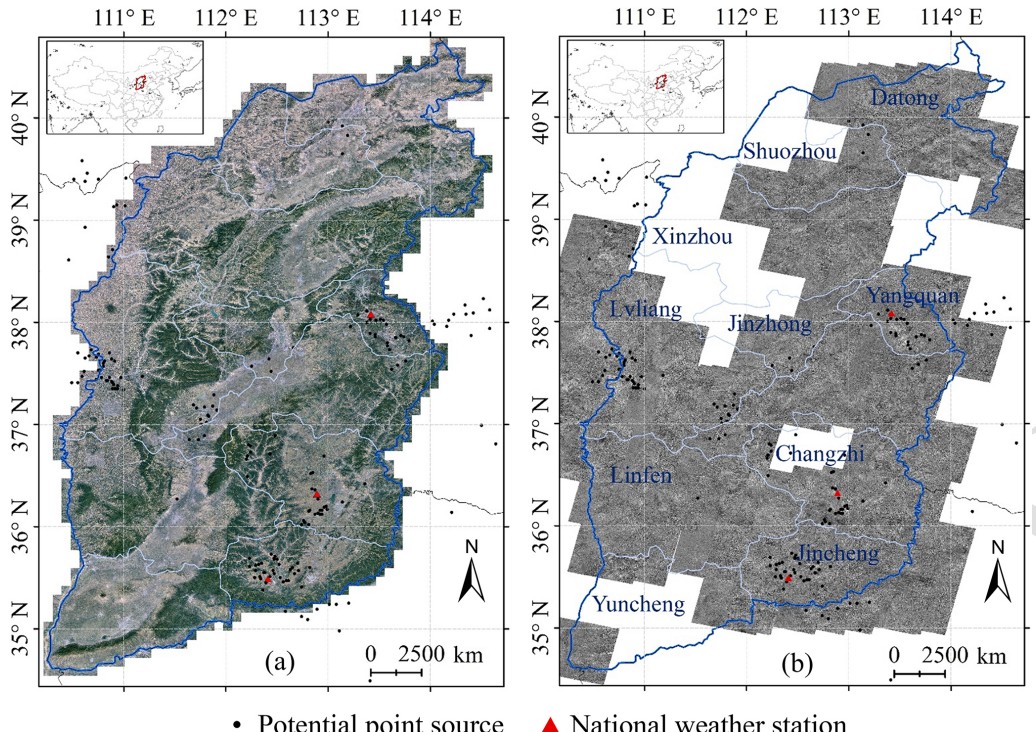

**Figure 1. (a)** The study area in Shanxi enclosed by the blue boundary, with the background image from © Google Maps. **(b)** Gaofen-5B observed scene images used for the CH$_4$ plume survey. The black dots represent the potential point sources detected by TROPOMI (Schuit et al., 2023). The red dots represent the three national weather stations for monitoring meteorological variables in Yangquan, Changzhi, and Jincheng that are used for the wind field comparison with ERA5 reanalysis (Sect. 4.3).

and FWHM change vary distinctly between different image scenes. It is therefore important to re-calibrate the spectra for all channels before further analysis using the observed spectra. While GF-5B AHSI imagery has been utilised in CH$_4$ point source detection experiments in various regions, estimation regarding its spectral offset and associated correction in FWHM has not yet been undertaken. In this study, we conduct this spectral calibration for the short-wave infrared (SWIR) channels from 2110 to 2455 nm of GF-5B/AHSI data (Guanter et al., 2009). The basic idea of the spectral calibration is to retrieve the wavelength shift and FWHM change that would lead to the best fit between observed GF-5B/AHSI spectra and the simulated spectra based on radiative transfer model. In practice, we used the forward radiative transfer model and optimal estimation method in GFIT3 (Zeng et al., 2021) to iteratively derive the spectral calibration parameters. Similar to Guanter et al. (2021), we applied the calibration to the averaged top-of-the-atmosphere radiance from all observations of each across-track detector and derive the wavelength shift and FWHM change. This calibration is repeated for all detectors and over all GF-5B/AHSI images. Eventually, the updated spectral centre wavelength and FWHM, instead of the nominal values, for all channels are used as inputs in the $\Delta$XCH$_4$ retrieval when the high-resolution CH$_4$ absorption spectra is convolved with Gaussian ILS.

### 3.1.2   Spectral matched filter for retrieving $\Delta$XCH$_4$

Spectral matched-filter method derives the $\Delta$XCH$_4$ by calculating the difference between the "polluted" spectra over a source region with background spectra of the ambient atmosphere and expressing the difference by the number of target absorption spectrum from one unit of XCH$_4$ (e.g. 1 ppm of XCH$_4$; Guanter et al., 2021). The retrieval using matched filter is depicted in Eq. (1):

$$\Delta\text{XCH}_4 = \left( (\boldsymbol{x} - \boldsymbol{\mu})^{\text{T}} {\textstyle\sum}^{-1} \boldsymbol{t} \right) \Big/ \left( \boldsymbol{t}^{\text{T}} {\textstyle\sum}^{-1} \boldsymbol{t} \right) , \tag{1}$$

where $\boldsymbol{x}$ denotes a vector of the observed SWIR hyperspectral spectra from a target pixel. In this study, the CH$_4$ strong absorption band (2110–2455 nm) is used; $\boldsymbol{\mu}$ and $\Sigma$ represent the mean and covariance of the SWIR hyperspectral observation over background regions, respectively. $\boldsymbol{t}$ is the target spectrum, representing the disturbance vector of SWIR hyperspectral due to enhanced XCH$_4$ relative to the background. It can be derived from an element-wise multiplication of $\boldsymbol{\mu}$ and the unit XCH$_4$ absorption spectrum $\kappa$, which is generated from GFIT3 (Zeng et al., 2021), as shown in Fig. 2, assuming a perturbation of 1 ppm XCH$_4$.

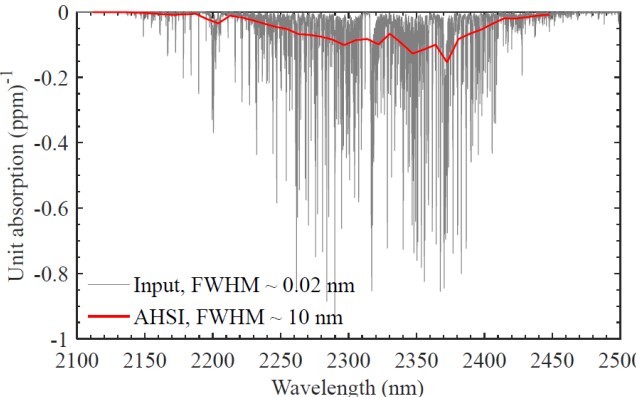

**Figure 2.** Example of unit XCH$_4$ absorption spectrum used as target signature in the matched-filter retrieval method. The high-resolution target signature (in grey) represents absorptivity induced by 1 ppm XCH$_4$ enhancement, which is calculated using GFIT3 (Zeng et al., 2021). The high-resolution absorptions are then convolved with a Gaussian ILS with nominal FWHM from GF-5B/AHSI to derive the spectra (in red) that can be compared with that observed from AHSI.

## 3.2 Identifying point source plumes from ΔXCH$_4$ maps

After data pre-processing, including spectral re-calibration and ΔXCH$_4$ retrieval, we implemented a geometric localisation to change the GF-5B/AHSI imagery index for row and column pixels to latitude and longitude under WGS84 projection. The detailed description of this geometric localisation is in Appendix A. Then, this study compares the ΔXCH$_4$ maps with high-resolution Google Earth imagery to visually inspect and preliminarily identify the CH$_4$ point source plumes. The identification criteria include: (1) high ΔXCH$_4$ values displaying plume characteristics, (2) the presence of ground facilities such as factories or pipelines representing potential CH$_4$ emission sources, and (3) plume distribution characteristics not caused by terrain features that may impact short-wave infrared strong absorption in surface features. Although wind conditions directly affect plume features, however, the reanalysis data (e.g. ERA5) of the wind direction may be very different from the plume structure. Several factors could contribute to this mismatch, including the temporal and spatial resolution of the reanalysis data, local topographical features, and microscale meteorological phenomena that are not fully captured by the reanalysis data. Therefore, this study temporarily refrains from utilising wind direction from ERA5 reanalysis as a direct criterion for point source identification.

## 3.3 Estimation of emission flux rates

### 3.3.1 Automatic segmentation of ΔXCH$_4$ plumes using flood-fill algorithm

The segmentation of ΔXCH$_4$ plumes in previous studies has often been manually drawn, which is a laborious and time-consuming process highly influenced by subjective human judgement, leading to possible bias in IME calculations. Hence, there is a need to introduce a statistically based, relatively objective, and easy-to-implement method for ΔXCH$_4$ plume segmentation. The flood-fill algorithm has been widely employed for segmenting and extracting continuous abnormal signals (He et al., 2018; Zscheischler et al., 2013), showing potential for ΔXCH$_4$ plume automatic segmentation. Specifically, this study uses statistical parameters, including ΔXCH$_4$ mean and 1 standard deviation, within the study area to segment and identify concentration-enhanced signals of ΔXCH$_4$. It employs the flood-fill algorithm to recognise abnormal pixels in the vicinity of eight directions, merging spatially connected pixels into a plume pattern by considering the spatial continuity of plumes. To carry out the flood-fill method in plume extraction, a background region needs to be defined to calculate the mean and standard deviation of ΔXCH$_4$, which set the basis for identifying anomalously high ΔXCH$_4$ in the plume relative to the background. In this study, for a specific plume, the origin is first pinpointed through visual interpretation. Then a background region is defined as a square using the source origin as the centre for calculating the mean ($\mu$) and standard deviation ($\sigma$) of ΔXCH$_4$. Finally, a threshold defined based on $\mu$ and $\sigma$ is used for the flood-fill algorithm to effectively segment the point source plume. The exact numbers for the background square length, $\mu$ and $\sigma$, are introduced in Sect. 3.3.3.

### 3.3.2 Estimation of CH$_4$ point source emission flux rates

For the emission flux rate estimation, this study employs the IME model (Eq. 2; Frankenberg et al., 2016; Varon et al., 2018; Guanter et al., 2021) to calculate the excess mass of CH$_4$ in the plumes relative to the background from the retrieved ΔXCH$_4$ plume maps. Then the emission flux rate ($Q$) is calculated using Eq. (3) with inputs of wind speed and the length of the plume. These equations are

$$\text{IME} = k \sum_{i=1}^{n_p} \Delta\text{XCH}_4(i) , \tag{2}$$

$$Q = ((\alpha \cdot U_{10} + \beta) \cdot \text{IME}) / L , \tag{3}$$

where $n_p$ denotes the number of pixels in the plume; ΔXCH$_4(i)$ represents the XCH$_4$ enhancement in pixel $i$; and $k$ is the scaling factor converting ΔXCH$_4$ from volume mixing ratio to mass based on Avogadro's law, considering the pixel resolution of GF-5B/AHSI to be 30 m. In Guanter

et al. (2021), $k$ is defined as $5.155 \times 10^{-3}\,\mathrm{kg\,ppb^{-1}}$, as derived from the surface pressure of one standard atmosphere. However, the average elevation of the identified plumes is 942.41 m (Fig. B1), whose surface pressure (900.64 hPa) is about 10 % less than one standard atmosphere. Therefore, we calculated a new $k$ based on the derived averaged surface pressure for all the identified plumes. The derived $k$ value ($4.565 \times 10^{-3}\,\mathrm{kg\,ppb^{-1}}$) is then used for estimating IME in this study. $Q$ denotes the point source emission rate, in unit of mass per unit time, obtained from IME calculation; ($\alpha \cdot U_{10} + \beta$) denotes the effective wind speed derived from wind speed at 10 m from ERA5 reanalysis; and $L$ is the plume length, defined as the square root of the plume mask area (Varon et al., 2018). $\alpha$ and $\beta$ can be determined through large-eddy simulation, based on the spatial resolution of satellite observation and $\Delta$XCH$_4$ retrieval accuracy from GF-5B/AHSI. In this study, we adopted the estimates ($\alpha = 0.37$ and $\beta = 0.64$) from Li et al. (2023) derived for the Changzhi region in Shanxi. Globally, the values of $\alpha$ and $\beta$ do not change significantly. For example, the values adopted for PRISMA (Guanter et al., 2021; Irakulis-Loitxate et al., 2021) were 0.34 and 0.44 and for GF-5B in the Permian basin (Li et al., 2023) were 0.38 and 0.41, respectively.

### 3.3.3 Estimation uncertainty in the point source emission flux rate

The uncertainty in point source emission flux rate typically involves two primary aspects: the IME calculation and wind speed. For the IME calculation based on the statistically driven method of flood-fill in plume extraction, the square background region and the threshold setting for plume enhancement segmentation are the main factors involved. Referring to the uncertainty assessment method by Cusworth et al. (2020), we first assess the uncertainty in the IME and then propagate the random errors in the IME and wind speed ($U_{10}$) to the flux rate $Q$, thereby evaluating the uncertainty in the estimated emission flux rate. In practice, for estimating IME and its uncertainty for a certain plume, we used six different background square lengths (from 12 to 24 km with an interval of 2.4 km) and six different segmentation thresholds (from $\mu + 0.45\sigma$ to $\mu + 0.55\sigma$, with an interval of $0.02\sigma$) for the flood-fill segmentation method (Fig. C1). Different values of $\mu$ and $\sigma$ are calculated for different background regions. This process enabled the extraction of 36 reasonable plume values, defining their mean and standard deviation as the IME estimation and its uncertainty, respectively. For the wind speed uncertainty, to be consistent with the previous study, we set it at 50 % for $U_{10}$ (Cusworth et al., 2020; Guanter et al., 2021). To further understand the uncertainty in the used wind fields, in Sect. 4.3, we have carried out an evaluation of wind speeds and wind directions from ERA5 reanalysis by comparison with observations from meteorological sites in Shanxi.

## 4 Results

### 4.1 Detection and estimation of emission flux rate for single CH$_4$ point source using GF-5B/AHSI

Figure 3 demonstrates the retrieval results of point sources $\Delta$XCH$_4$ based on multiple capturing of the same point source using GF-5B/AHSI from January 2022 to August 2023. Under different emission flux rates and wind conditions, the emission plumes exhibited various characteristics. Six observations occurred during the winter–spring seasons (Fig. 3b–g), showing $\Delta$XCH$_4$ plumes spreading northeastward, while the observation in summer (Fig. 3h) displayed a plume drifting northwestward. An intriguing aspect is the occurrence of two repeated observations of the same point source within an 8 s interval (Fig. 3d, e and f, g). Theoretically, CH$_4$ emissions from the same point source within an 8 s interval should exhibit very similar patterns. However, using the full-scene image as the background region for the background spectra calculation for each plume, similar to previous studies, the $\Delta$XCH$_4$ of the plumes from the same point source showed large differences, especially for Fig. 3f and g. The notable difference primarily arises from the different background used, suggesting the importance of selecting appropriate background regions. Note that the difference may also be slightly caused by the different signal-to-noise ratio (SNR), as the plumes appear at different locations in the imaging scene. The plumes appear at the bottom of the scene in Fig. D1f and at the top in Fig. D1g. They may be observed by different detectors with different SNR of the instrument that affect the detection accuracy of the plumes.

Based on the $\Delta$XCH$_4$ retrieval and the flood-fill plume segmentation method, we obtained the plume characteristics and emission flux rate of the seven detections, as shown in Fig. 4. The results indicate the following: (1) differences exist between the extracted plume features and visual segmentation. For instance, in Fig. 4c, the elaborate plume automatically extracted using flood-fill would be challenged for manually drawing; (2) the point source emission flux rate varies between $2834.79 \pm 1330.98\,\mathrm{kg\,h^{-1}}$ in Fig. 4e and $6678.66 \pm 2316.49\,\mathrm{kg\,h^{-1}}$ in Fig. 4g (excluding incomplete observations in Fig. 4d). Among these observations, four fall within a similar range between 2834.79 and $3247.22\,\mathrm{kg\,h^{-1}}$; (3) the uncertainty in the IME ranges from 5.07 % to 66.38 %, with the majority being below 30 %, which is lower than the uncertainty caused by wind speed ($\sim$ 50 %) in the emission flux rate calculation. (4) Significant differences are evident in the plumes from adjacent detections of the same point source (e.g. Fig. 4f and g), indicating the different backgrounds chosen for different imagery scenes are not optimal to monitoring the same emission plumes.

In order to eliminate the impact of background selection on estimating emission flux rate from the same point source, this study conducted a $\Delta$XCH$_4$ retrieval experi-

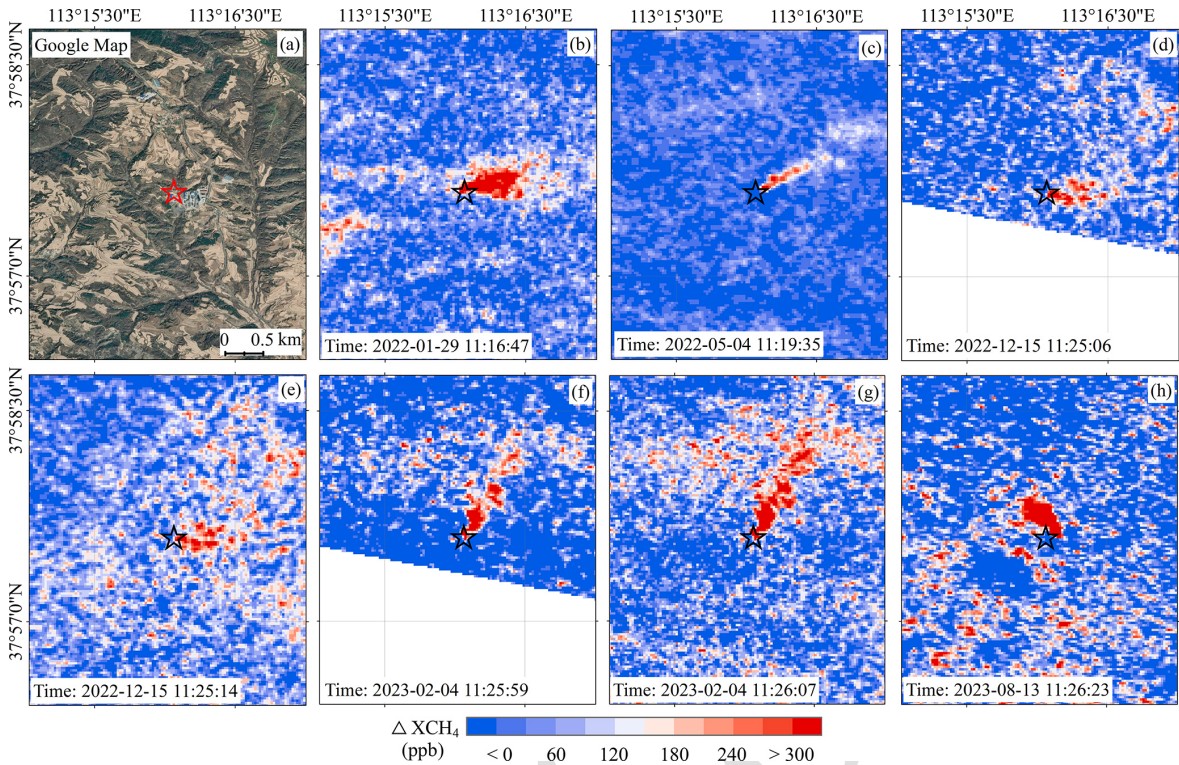

**Figure 3.** Example of $\Delta XCH_4$ retrievals from one typical point source with multiple overpasses by GF-5B/AHSI, with its origin (lat $37°57'36''$, long $113°16'04''$) marked with a red/black star. The detected plumes from the seven overpasses shown in panels **(b)**–**(h)**. The observation times (in UTC + 8, standard Beijing time) are shown for each plume event, which are close to local time. The background image in panel **(a)** is adopted from © Google Maps.

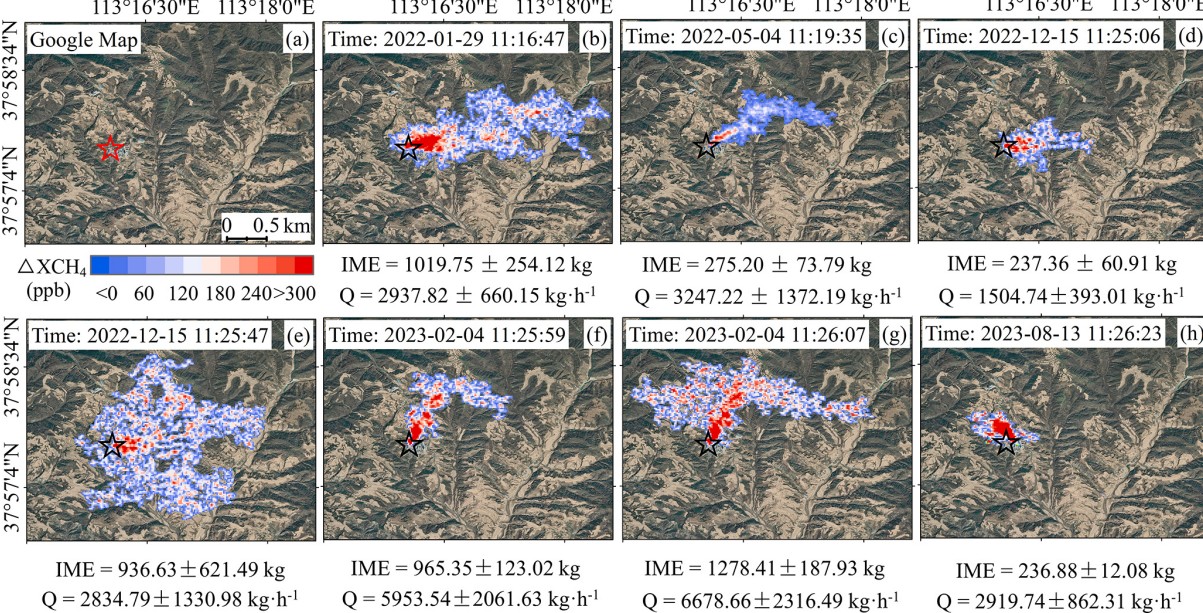

**Figure 4.** Examples of the extracted $CH_4$ point source, marked with red/black star (lat $37°57'36''$, long $113°16'04''$), with the plume using flood-fill method based on the retrieved $\Delta XCH_4$ maps, as shown in Fig. 3, from a single point source with multiple overpasses by GF-5B/AHSI. The plume mass from IME model and the estimated emission flux rates are also indicated at the bottom of each map. The observation time in Beijing time is shown for each plume event. All background images in panels **(a)**–**(h)** are from © Google Maps.

ment using overlapping area in the imagery maps of Fig. 3f and g as the new background. The results based on the new backgrounds shown in Fig. 5 demonstrate highly similar $\Delta$XCH$_4$ plume features between the two observations that are 8 s apart (Fig. 5a and b). The extracted plume distribution and emission flux rate calculations shown in Fig. 5c and d are almost identical. The integrated enhanced masses were $1001.59 \pm 13.98$ and $1046.86 \pm 15.20$ kg, respectively, with emission flux rates of $5477.44 \pm 1839.08$ and $5730.52 \pm 1925.58$ kg h$^{-1}$. This reduced estimation discrepancy between the two by $485.11$ kg h$^{-1}$, which is about 8.5 % of the emission flux rate. Therefore, for the calculation of emissions from all plumes in Shanxi, we adopted a two-step approach to identify CH$_4$ plumes and estimate emissions. In step 1, the whole image is used to calculate $\Delta$XCH$_4$ and identify plumes; in step 2, when implementing the flood-fill method with the strategy of selecting background regions as described in Sect. 3.3.3, the $\Delta$XCH$_4$ is re-calculated using the same background regions for the flood-fill method. In other words, the chosen background regions are used for calculating $\Delta$XCH$_4$ using Eq. (1), segmenting plumes using flood-fill method, and estimating IME using Eq. (2).

## 4.2 Spatial distribution of point sources and their emission rates in Shanxi

Based on the methods described above for estimating the CH$_4$ emission flux rate of point sources, we conducted a survey of all detectable point source emissions using all available imagery of GF-5B/AHSI from 2021 to 2023. In total, 93 point source plumes were identified. After averaging repetitive observations over the same point sources, a total of 32 point sources were identified, and their spatial distribution is depicted in Fig. 6. Figure 6a–i exhibit typical plume extraction results around three typical cities of Yangquan, Changzhi, and Jincheng. The emission flux rates range from $2147.08 \pm 427.42$ to $9198.03 \pm 2059.18$ kg h$^{-1}$. This result demonstrates a reasonably good consistency between the spatial locations of the actual CH$_4$ emission point sources identified in this study (red dots in Fig. 6) and those extracted based on TROPOMI data (black dots in Fig. 6), which are primarily concentrated around the three cities of Yangquan, Changzhi, and Jincheng. Given its high spatial resolution, the spatial locations derived from GF-5B/AHSI are expected to be more accurate. We found that the number of identified point sources is much fewer than those extracted from TROPOMI. This is primarily attributed to the much denser observations with daily global coverage and its different overpass time from TROPOMI ($\sim$ 13:30 LT, local time). In addition, the high-resolution of the $\Delta$XCH$_4$ retrieval results helped eliminate false positive signals due to surface interference elements like photovoltaic panels (further details are discussed in Sect. 4.3.2) and greenhouse cultivation structures that are ubiquitous in Shanxi. Driven by wind speed and topography, different plumes from various point sources

show distinctly varying dispersion distances, ranging from less than 1.0 km (e.g. Fig. 6h) to 5.0 km (e.g. Fig. 6d). In the calculation of emission flux rate, the plume length plays an important role in the background region selection, which affects both the inversion of $\Delta$XCH$_4$ (background spectra) and plume segmentation using the flood-fill algorithm (statistical parameters).

We further conducted IME calculations and emission flux rate estimations for the 93 plumes extracted from GF-5B/AHSI (Fig. 7a and c). Additionally, based on multiple observations (from two to eight times) of the same point source, we provided the highest and lowest emission flux rates and IME for the same point source (Fig. 7b and d). The survey results revealed a diverse range of point source emission flux rates, varying from $761.78 \pm 185.00$ (minimum) to $12\,729.12 \pm 4658.13$ kg h$^{-1}$ (maximum), with an average of approximately $4040.30$ kg h$^{-1}$. The range of IME from point source emissions spans from $33.58 \pm 6.27$ (minimum) to $6587.50 \pm 1925.31$ kg (maximum). This discrepancy in the range of IME and the emission flux rates is indeed notable and can be attributed to the variability in wind conditions. Moreover, assuming a 50 % uncertainty in $U_{10}$ (wind speed at 10 m), in the uncertainty calculation of Q (emission flux rate), the impact of wind speed and IME uncertainties accounts for approximately 84.84 % and 15.16 %, respectively. While it is evident that wind speed is the predominant factor contributing to the uncertainty in estimating CH$_4$ point source emissions, it is crucial to acknowledge the significance of IME-related uncertainties. Because the wind uncertainty could be decreased significantly with accurate weather observations from high-density sites from China Meteorological Administration in theory. In addition, we should acknowledge that low wind speeds may also lead to uncertain dispersion of methane plumes, potentially biasing emission estimates, which need deeper analysis in future studies.

Furthermore, multiple observations of the same point source indicate significant variations in CH$_4$ emissions over time. The difference is as large as $10\,204.71$ kg h$^{-1}$, which is about 7 times the amount between the maximum and the minimum, as shown in Fig. 7d. This difference suggests that a single observation does not adequately represent the overall or averaged emission scenario for any point source. Because the specific emission law of each emission point is unclear, more coal mine emission time series detection experiments (Qin et al., 2023) are needed for the overall emission rate evaluation. Although, Y. Chen et al. (2022) used high-density (26 292 active wells) and highly repeated (115 flight days) measurements from an aerial instrument to quantify methane emissions from the whole regional study area of the New Mexican Permian Basin with the persistence-averaged method. The persistent emission rate from a single point source was calculated with the emission detection probability derived from highly repeated observations. In this study, this may not be feasible because the observations are too few to calculate the possibility of emission detection. As the emis-

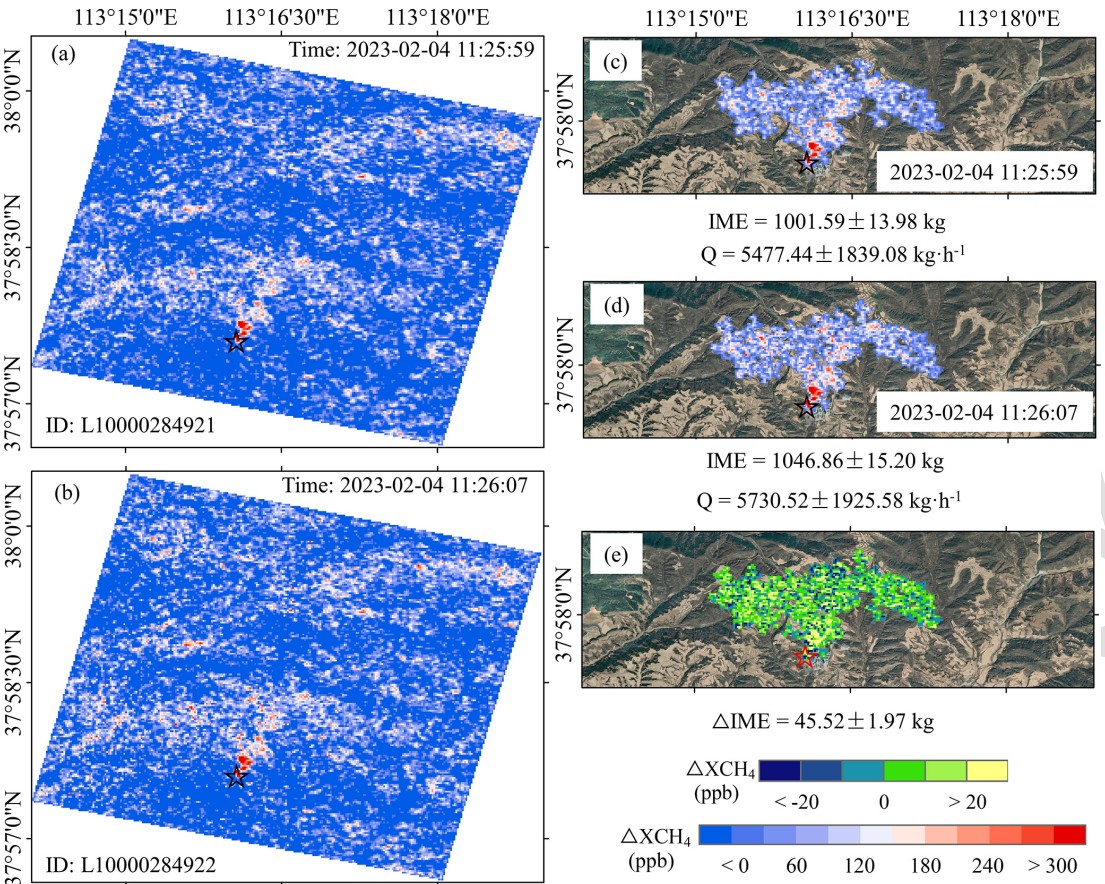

**Figure 5.** $\triangle XCH_4$ retrievals from GF-5B/AHSI observations that are 8 s apart in panels **(a)** and **(b)** over the same point source and marked with red/black star (lat $37°57'36''$, long $113°16'04''$). The retrievals are carried out using the same background region. The corresponding IME values and emission flux rates ($Q$) based on the extracted $\triangle XCH_4$ maps are shown in panels **(c)** and **(d)**, respectively. The difference between the two IME values is shown in panel **(e)**. All background images in panels **(c)**–**(e)** are from © Google Maps.

sion rates of the detected plume in this study (Fig. 7), the detection limit based on GF-5B AHSI and our current method could be $761.78 \pm 185.00 \, \text{kg h}^{-1}$. However, there might be some smaller point source emissions in Shanxi, which could be detected with the inversion method improved.

### 4.3 Improvements on $\triangle XCH_4$ retrieval and emission flux rate estimation

#### 4.3.1 Spectral calibration of GF-5B/AHSI observations

The impact of the wavelength shift and changes in FWHM of the spectral observations from GF-5B/AHSI on deriving $\triangle XCH_4$ is demonstrated in Figs. 8 and 9. Figure 8 illustrates an example of the cross-track pixel variations in the estimated centre wavelength in Fig. 8a and FWHM in Fig. 8b in a single-scene image collected on 29 January 2022. The results reveal the distinct deviations from the nominal centre wavelength and FWHM among different track pixels during satellite imaging. Figure 8c displays the $\triangle XCH_4$ of the corrected image, capturing plumes seen in Figs. 3b and 6c, d,

among others. Figure 8d and e show the evident striping differences and spectral calibration's impact on calculating $\triangle XCH_4$ of individual plume. The difference can reach up to 100 ppb. To further assess the spectral calibration's influence on $CH_4$ point source estimation, this study analysed the shift in centre wavelength and changes in FWHM in 111 representative scenes with potential point source emissions using GF-5B/AHSI, as shown in Fig. 9a and b. The results indicate that the average shift in centre wavelength of GF-5B/AHSI is approximately $-0.05$ nm, mostly ranging between $-0.2$ and $0.1$ nm. The ratio of change in FWHM averages around $1.1$, predominantly falling between $1.0$ and $1.25$ times (between 0 and 2.13 nm). Furthermore, the study evaluated the impact of spectral shift and FWHM change on the estimation of point source emission flux rate, as shown in Fig. 9c and d. The results indicate that the caused difference in the point source emission flux rate ranges from $0.43$ to $500.96 \, \text{kg h}^{-1}$. The average percentage of change is $(1.78 \pm 1.39)\%$. The maximum difference reaches up to about $5.0\%$. By considering the shift in central wavelength and change in FWHM in the spectral

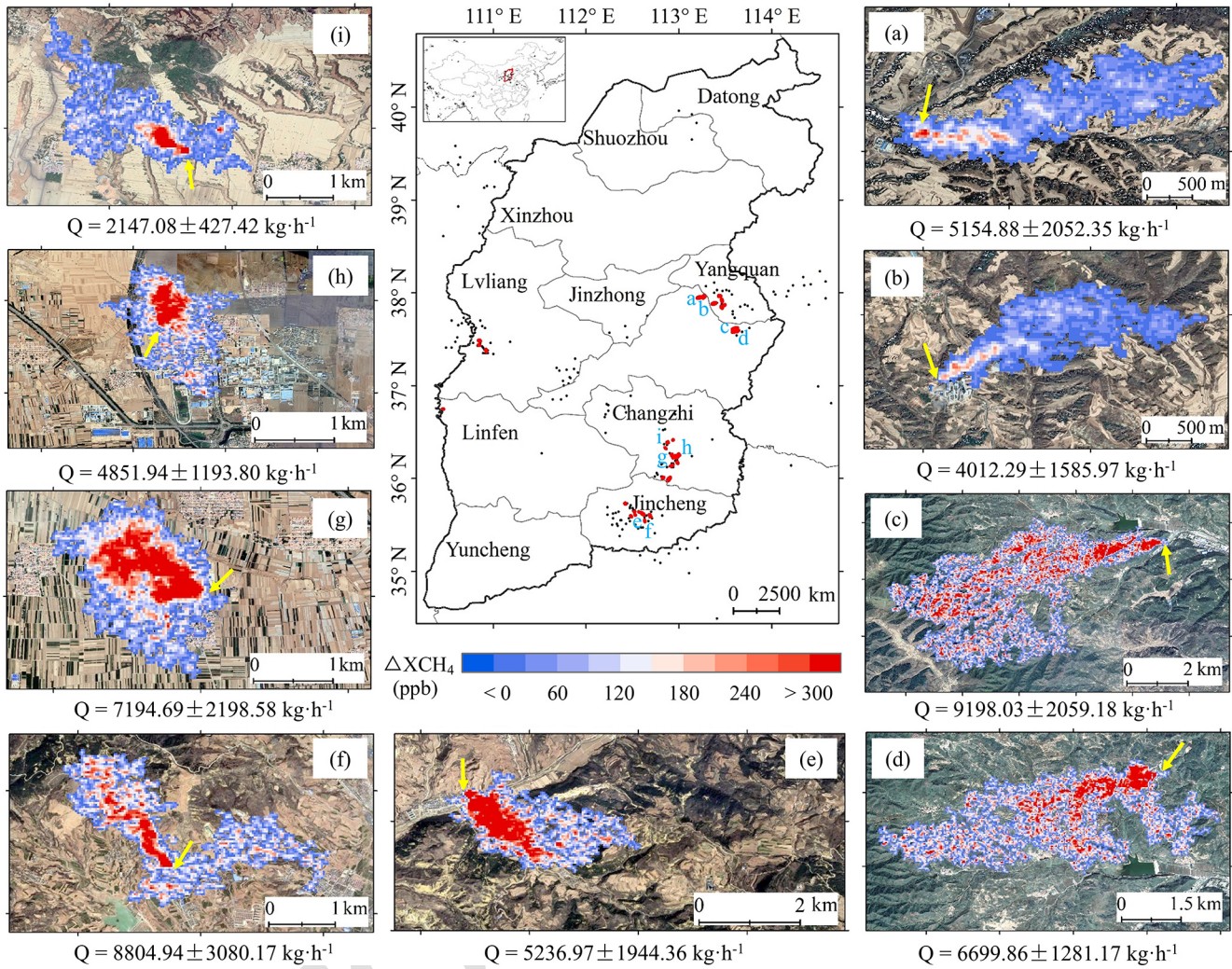

**Figure 6.** The spatial distribution of the identified CH$_4$ plumes (in red dots; in total 93) in Shanxi using GF-5B/AHSI observations, as shown in the centre panel. The black dots represent the potential point sources detected by TROPOMI (Schuit et al., 2023). CH$_4$ plumes **(a–i)** are examples of the identified $\triangle$XCH$_4$ plumes in Shanxi, and the yellow arrow points to the origins of the identified point sources. All background images in panels **(a)**–**(i)** are adopted from © Google Maps.

observations, it exhibits a potential to reduce the uncertainty in the XCH$_4$ emission rate estimation using GF-5B/AHSI.

### 4.3.2   Impact of heterogeneous surface features

Complex surface features significantly affect the identification of suspected point sources based on $\triangle$XCH$_4$ maps and the derivation of point source emissions. In this study, we originally observed 219 instances of 113 suspicious point sources. In a more refined identification of these sources, we cross-checked and confirmed their positions using $\triangle$XCH$_4$ retrievals from GF-5B/AHSI against high-resolution Google Earth imagery. Our findings revealed that the identification of point sources was significantly affected by the complex surfaces that exhibit strong SWIR absorption similar to CH$_4$ and therefore result in false positive signal. Notably, the ar-

ray of solar panels that is widespread in Shanxi is the primary disruptor of the spectral matched-filter retrieval method. An example of solar panel arrays is shown in Fig. 10. Moreover, we found that surface features such as greenhouse structures, certain buildings, waterbodies with plume-like distributions, and moist cultivated lands (like paddy fields) also generated noticeable high-value $\triangle$XCH$_4$ interference signals. Therefore, in CH$_4$ point source detection using GF-5B image, it is essential to consider combining with high-resolution images to filter out false positive signals.

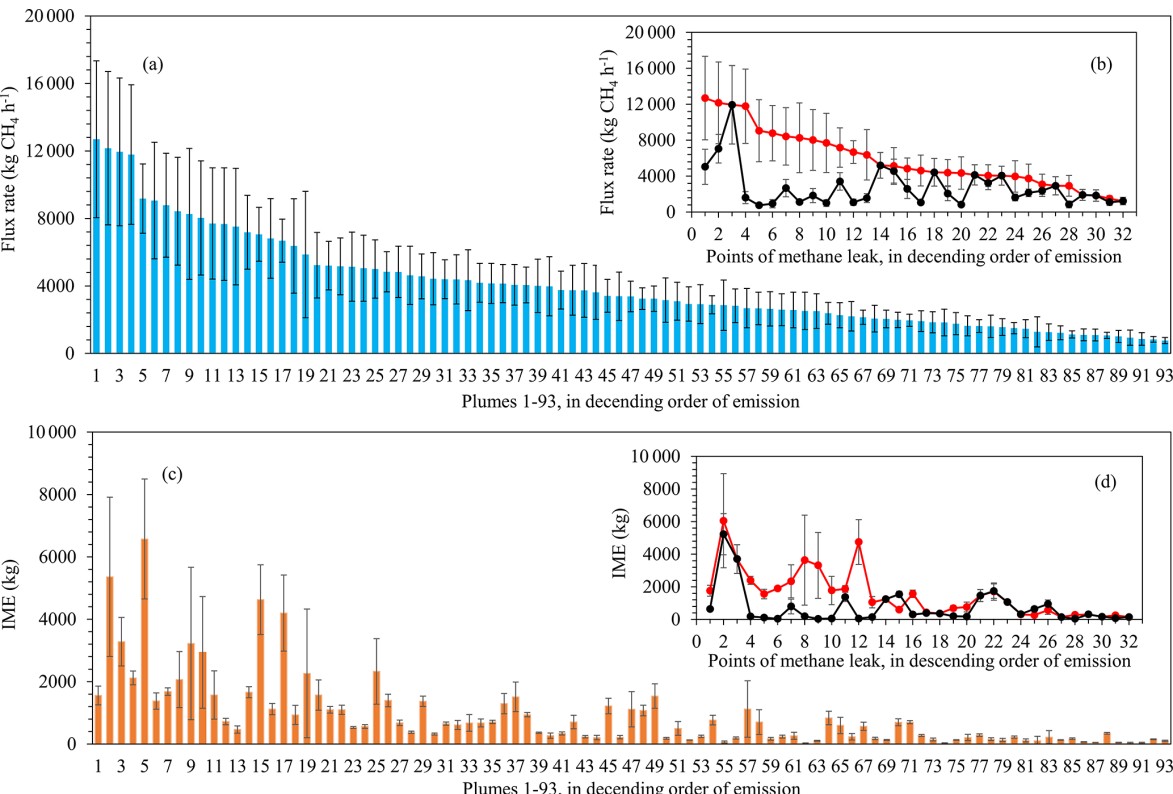

**Figure 7. (a)** $CH_4$ emission flux rate from point source plumes nos. 1–93 in descending order of emissions, with the error bars representing the estimation uncertainty. **(b)** The maximum and minimum emission flux rates for each point source with more than two observations. **(c)** The corresponding IME estimates for plume nos. 1–93, following the order in panel **(a)**. **(d)** The maximum of the minimum emission flux rates for each point source with more than two observations.

### 4.3.3 Evaluating wind fields from ERA5 reanalysis using observations from meteorological stations in Shanxi

Wind fields, including wind speed and direction, are the primary drivers of uncertainty in estimating point source emissions, especially in plume segmentation and flux rate calculations. For plume segmentation, instead of visual interpretation, this study introduces the flood-fill method. Instead of searching in eight directions in the current study, accurate wind direction information could enable us to precisely narrow down the flood-fill search directions, thereby removing abnormal signals from non-point source emissions, enhancing the reliability of plume segmentation. In the emission flux rate estimation, aligning with previous studies, this study defined an uncertainty in ERA5 wind speed as 50 %, thus leading to a significant uncertainty in the estimated emission rate. To evaluate the uncertainty in the wind fields from ERA5 reanalysis, which is widely used in many previous studies, this study compared them with data from three ground-based meteorological sites in Shanxi over the concentrated point source areas (Fig. 11). The comparison results indicate that from 2021 to 2023, the overall bias in the ERA5 wind speed

was approximately $1.30\,\mathrm{m\,s^{-1}}$, which is close to 100 % of bias on average. It has been recognised that the wind speed should be in a moderate range to allow detectable plumes from space. A wind speed that is too small may hamper the development of the plume, while a wind speed that is too large may dilute the plume. It is observed in our cases that the wind speeds roughly fall within 0.5 to $2.5\,\mathrm{m\,s^{-1}}$ for most days with detectable point source plumes. If we assume this is the suitable wind speed range for satellite detection, as shown in black dots in the upper panel of Fig. 11, the deviation is about $0.45$–$0.54\,\mathrm{m\,s^{-1}}$, which is close to about 50 % of the wind speed from ERA5. This uncertainty is consistent with the assumption of wind speed uncertainty (50 %) in this study. In terms of wind direction, there are significant differences between ERA5 and the observations from meteorological sites. While ERA5 reanalysis data (at a height of 10 m) show relatively constant wind direction, the measurements of wind direction from meteorological stations show a much larger range. This discrepancy indicates significant deviations between ERA5 reanalysis wind fields and actual wind conditions, challenging their direct application in point source plume identification and emission estimation. Consequently, leveraging high-density and high-precision meteo-

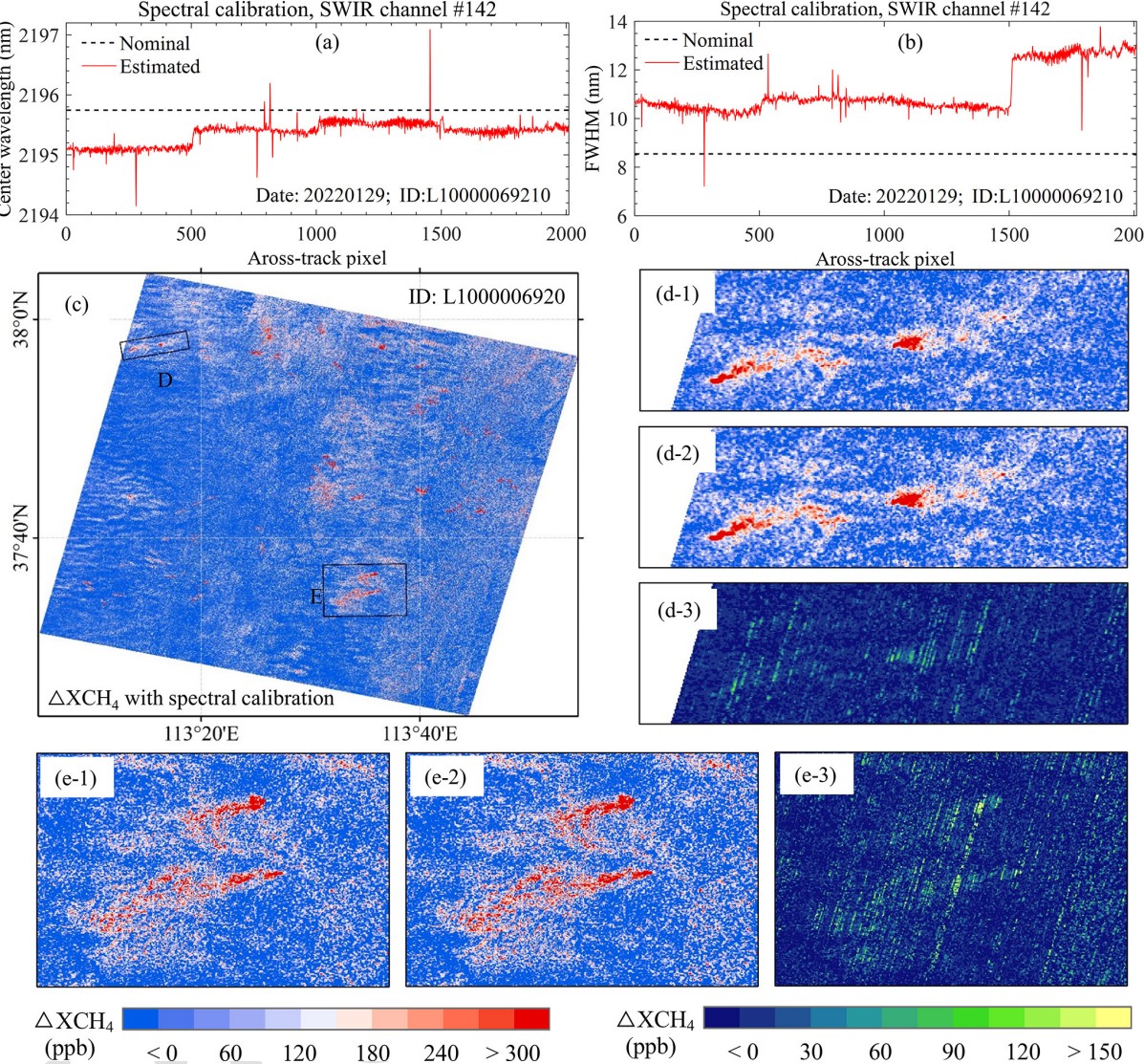

**Figure 8.** Example of the shift in the centre wavelength and FWHM change for across-track pixels of channel no. 142 from GF-5B/AHSI SWIR imagery and their impacts for $\Delta XCH_4$ retrieval. Panel **(a)** shows the shift in the centre wavelength for across-track pixels. Panel **(b)** shows the FWHM variation ratio for across-track pixels. Panel **(c)** shows the $\Delta XCH_4$ retrieval of a single image with inputs of updated spectral calibration parameters. Panels **(d)** and **(e)** are the comparison of zoomed-in plumes with and without the inputs of updated spectral calibration parameters, in which panels (d-1) and (e-1) are results without calibration, panels (d-2) and (e-2) are results with calibration, and panels (d-3) and (e-3) are the corresponding difference in $\Delta XCH_4$ retrieval.

rological observations from automatic meteorological monitoring stations, especially over regions with complex surface properties, could reduce the uncertainty and enhance the accuracy of satellite-based detection and estimation of $CH_4$ point source emissions.

A flat 50 % wind error could underestimate uncertainty for slow winds and overestimate uncertainty for fast winds. Therefore, we carried out an evaluation of the plume emission uncertainty using the absolute wind error (1.297 m s$^{-1}$ on average) estimated by comparing wind speeds from EAR5 and local meteorological stations in Shanxi. The results of

$CH_4$ flux rates and their uncertainty are shown in Fig. E1. As we expected, the uncertainty in the flux decreased/increased at high/low wind speed, respectively. In addition, the impact of wind speed uncertainties accounts for approximately 86.31 %, which is similar to the previous result based on a flat 50 % wind error. This result supports the fact that wind speed remains the dominant factor contributing to the uncertainty in estimating $CH_4$ point source emissions.

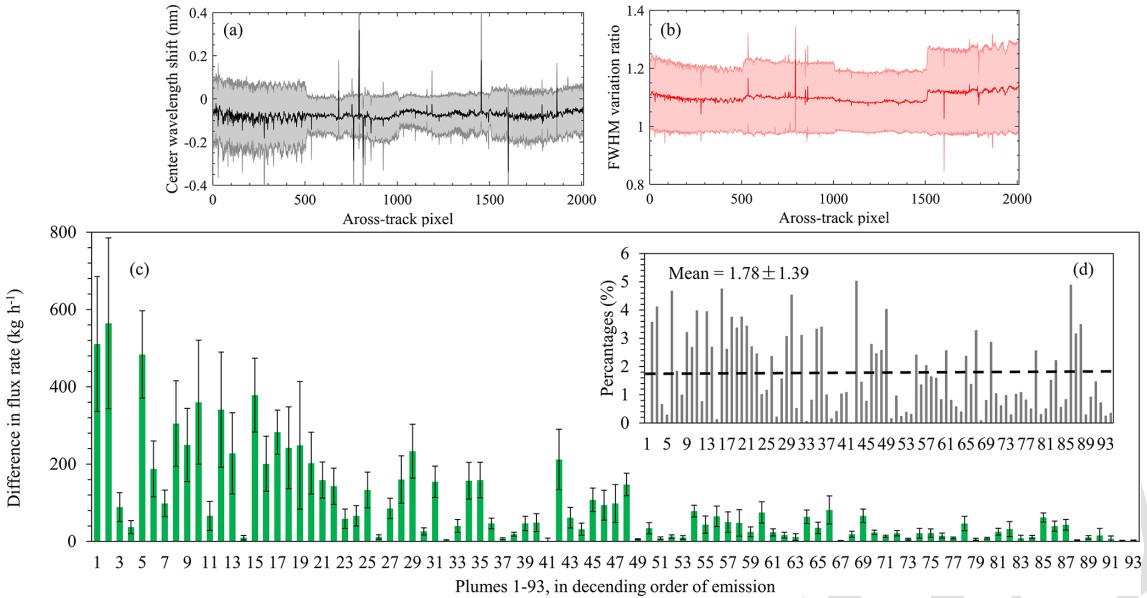

**Figure 9. (a)** Statistics of the shift in centre wavelength and **(b)** FWHM variation ratio of all 111 GF-5B/AHSI SWIR images with potential $CH_4$ point sources. **(c)** The difference in the estimations of emission flux rates and **(d)** the corresponding difference in percentages for all detected $CH_4$ plumes shown in Figs. 6 and 7.

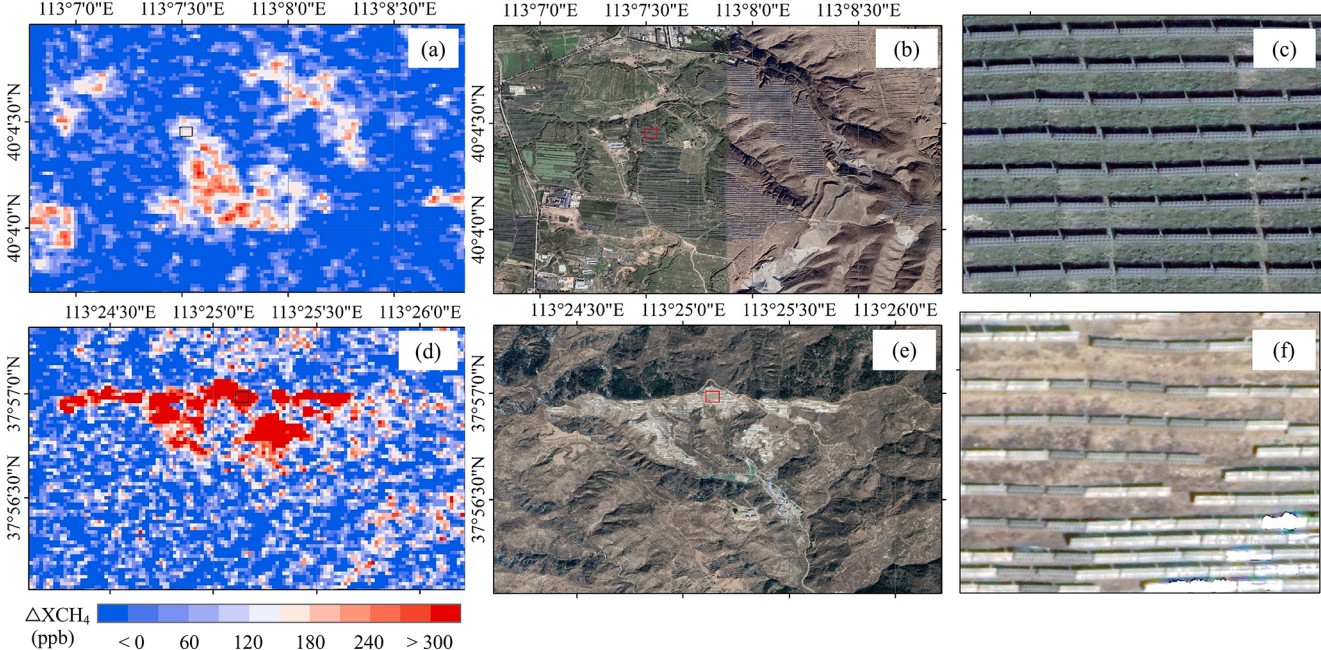

**Figure 10.** Examples of the impact of array of solar panels, which generates false positive signals, on the $\Delta XCH_4$ retrieval in Shanxi. $\Delta XCH_4$ retrievals with high values are similar to plume shapes in panels **(a)** and **(d)**. The false positive signals are caused by the similar patterns of solar panel arrays, which can be seen from high resolution of the ©Google maps in panels **(b)** and **(e)**. Zoomed-in details of solar panels in the red boxes in panels **(b)** and **(e)** can be found in panels **(c)** and **(f)**, respectively.

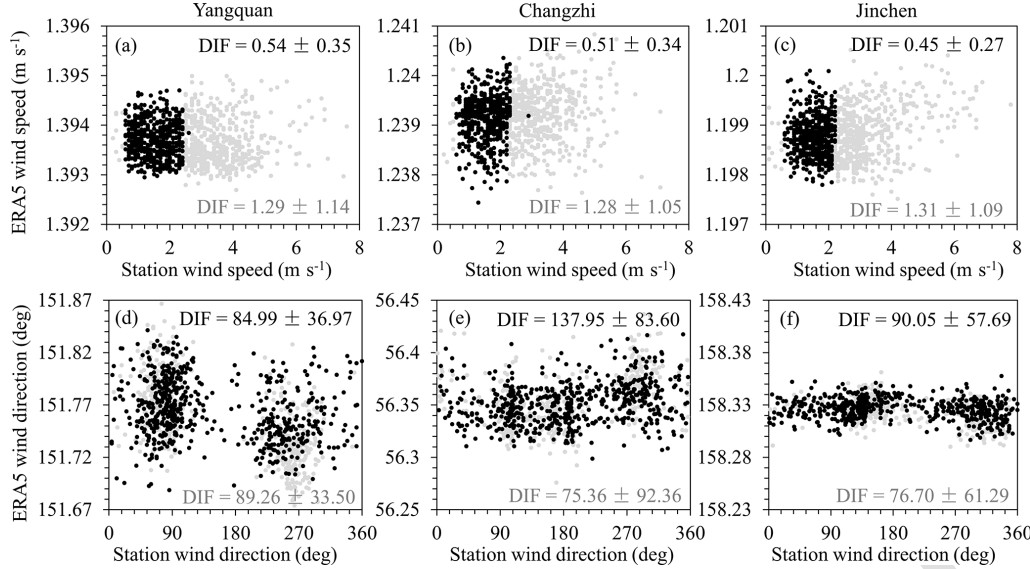

**Figure 11.** Comparison of wind speeds **(a–c)** and directions **(d–f)** between ERA5 reanalysis and meteorological stations located in Yangquan, Changzhi, and Jincheng cities in Shanxi, as indicated in Fig. 1. Wind field data from 2021 to 2023 are extracted in the daytime and correspond to the GF-5B overpass time. Black dot data are selected according to the wind speed range of 0.5 to 2.5 m s$^{-1}$.

## 5    Summary

In this study, we conducted a survey of CH$_4$ point sources emissions from coal mines in Shanxi, China, using hyperspectral observations of GF-5B/AHSI. We first carried out the spectral calibration based on the estimates of the across-track changes in channel centre wavelength and FWHM, which are approximately $-0.05$ nm and 10 %, respectively. We adopted the widely used matched-filter method to calculate the enhancement $\Delta$XCH$_4$. Based on the enhancement, the emission plumes are extracted using the fill-flood method, which is an automated plume segmentation method. The emission flux rate and the associated uncertainty are eventually estimated using IME method. Our results show that the errors caused by spectral calibration (wavelength shift and FWHM change) and the selection of different background can reach up to 5.0 % and 8.5 %, respectively. Simultaneously, this study presents the spatial distribution and emission flux rates of 32 point sources and 93 observed plumes in Shanxi province from 2021 to 2023. The findings indicate that coal mine sources in Shanxi are primarily located around Yangquan, Changzhi, and Jincheng areas, with plume emission flux rates ranging from $761.78 \pm 185.00$ (the minimum) to $12\,729.12 \pm 4658.13$ kg h$^{-1}$ (the maximum). Multiple repeated observations show significant differences in emission flux rates from the same source. The difference can reach to $10\,204.71$ kg h$^{-1}$ with a different by a factor of more than 7 times between the maximum and the minimum, indicating that a single overpass observation cannot represent the overall emissions of the point source. This study highlights that wind speed remains the primary factor con-tributing to uncertainty in point source emission estimation (approximately 84.84 %), yet the uncertainty in the IME (approximately 15.16 %) is also important.

It is important to note that the plume shapes detected based solely on the $\Delta$XCH$_4$ maps contains false positive signals due to surface interference. The strong absorption in SWIR by certain surface types significantly disrupts point source detection and flux rate emissions. In the future, a fusion of hyperspectral spectra and multispectral image with high spatial resolution could effectively filter out false positive signals and remove surface covering interference. In addition, the uncertainty in the wind field data remains significant sources of uncertainty in CH$_4$ point source emission flux rate estimation. From the evaluation of the accuracy of wind fields in ECMWF ERA5 reanalysis by comparing with ground-based meteorological station, we found a large discrepancy, especially in wind directions. For regions with complex terrain like Shanxi, incorporating local meteorological measurements into the detection of the CH$_4$ point source are important to achieve high accuracy.

## Appendix A:    Geometric localisation of GF-5B/AHSI images

The identification of CH$_4$ point sources using high-resolution satellite imagery is closely linked to land cover, while the accurate calculation of $\Delta$XCH$_4$ is significantly affected by spectral differences in the background within the study area. Hence, precise geometric localisation (Eqs. 4–6) of the GF-5B satellite images is crucial. The retrieval of $\Delta$XCH$_4$ involves both forward and inverse computations of the ratio-

nal polynomial coefficients (RPCs) in high-resolution imagery (Q. Liu et al., 2019). The forward-computation entails transforming the row and column indices ($Row_i$, $Col_i$) of the image data into geographical coordinates ($Lat_i$, $Long_i$), aiding in detecting and identifying $\Delta XCH_4$ point sources. Conversely, the reverse computation aims to optimise background concentration calculations by transforming detected point source geographical coordinates back to the image's row and column indices.

$$\begin{cases} Row_i = F_a(U_i, V_i, W)/F_b(U_i, V_i, W_i) \\ Col_i = F_c(U_i, V_i, W)/F_d(U_i, V_i, W_i), \end{cases} \quad (A1)$$

$$\begin{aligned} F_a(U, V, W) =\ & a_1 + a_2 V + a_3 U + a_4 W + a_5 V U \\ & + a_6 V W + a_7 U W + a_8 V^2 + a_9 U^2 \\ & + a_{10} W^2 + a_{11} U V W + a_{12} V^3 \\ & + a_{13} V U^2 + a_{14} V W^2 + a_{15} V^2 U \\ & + a_{16} V^3 + a_{17} U W^2 + a_{18} V^2 W \\ & + a_{19} U^2 W + a_{20} W^3, \end{aligned} \quad (A2)$$

$$\begin{cases} U_i = (Lat_i - Lat\_off)/Lat\_scale \\ V_i = (Lon_i - Lon\_off)/Lon\_scale \\ W_i = (Height_i - Heigh\_off)/Heigh\_scale, \end{cases} \quad (A3)$$

where $a_1 \ldots a_{20}$, $b_1 \ldots b_{20}$, $c_1 \ldots c_{20}$, $d_1 \ldots d_{20}$, Lat_off, Lat_scale, Lon_off, Lon_scale, Heigh_off, and Heigh_scale are rational polynomial coefficients (RPCs) which can be obtained from the incidental data of the GF-5B images.

## Appendix B: Statistics of the digital elevation for the origin from GF-5B AHSI detected plumes

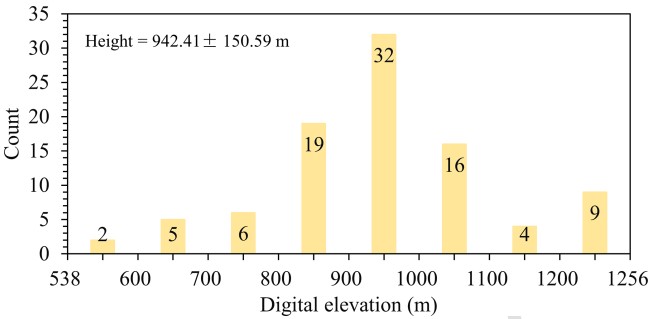

**Figure B1.** Histogram of the elevation for the detected plumes in Shanxi. The elevation data are from the DEM shown in Sect. 2.3.

**Appendix C: Examples of the integrated mass enhancement (IME) uncertainty calculation**

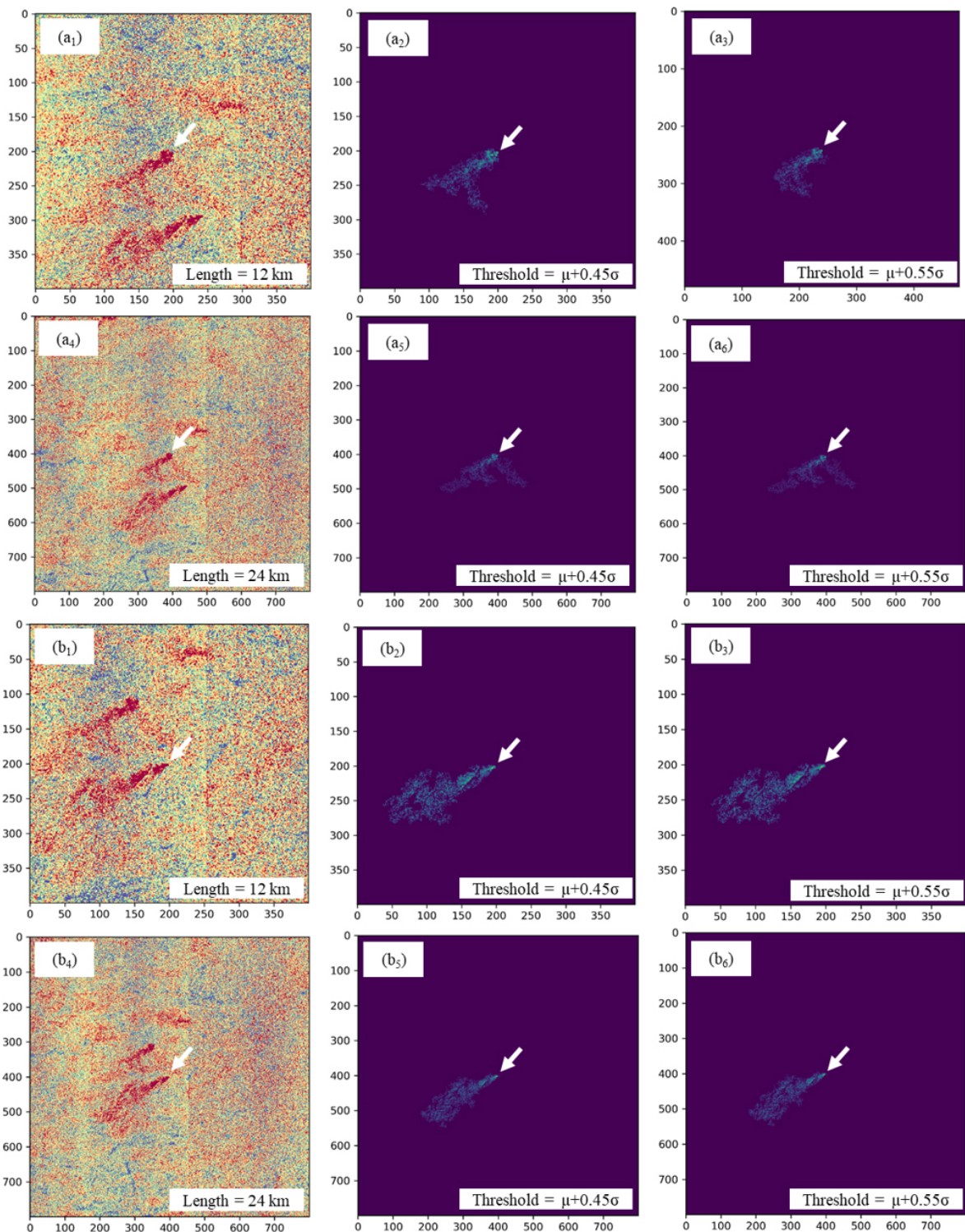

**Figure C1.** Examples of plume segmentation in the flood-fill method using different lengths for the background square and different segmentation thresholds. Two plumes are given in $a_1$–$a_6$ and $b_1$–$b_6$ as examples, in which $a_1$–$a_3$ and $b_1$–$b_3$ are for the length of 12 km and $a_4$–$a_6$ and $b_4$–$b_6$ are for the length of 24 km. Two different thresholds, $\mu + 0.45\sigma$ and $\mu + 0.55\sigma$, are given for the two plume examples.

## Appendix D: An example of the same plume target inside the nearest two full images

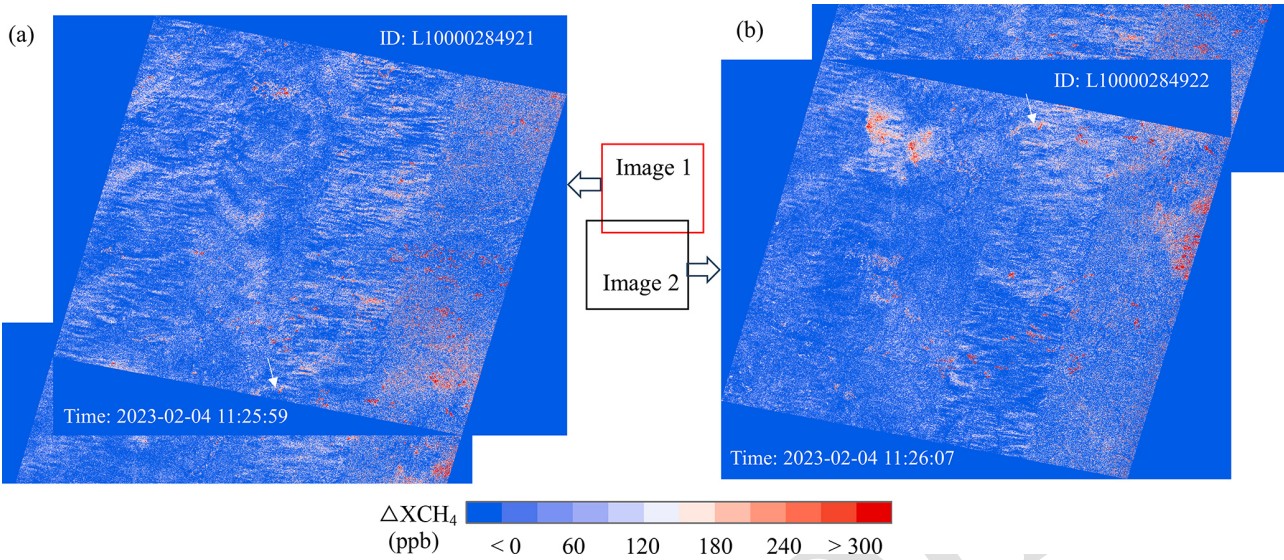

**Figure D1.** The full images of $\triangle XCH_4$ for Fig. 3f **(a)** and Fig. 3g **(b)**. The plume target (indicated by the white arrow) appears in the overlapping region of the two images.

## Appendix E: CH$_4$ emission flux rates from point source plumes in Shanxi, with an absolute wind speed uncertainty estimated by comparing wind speeds from EAR5 and local meteorological stations

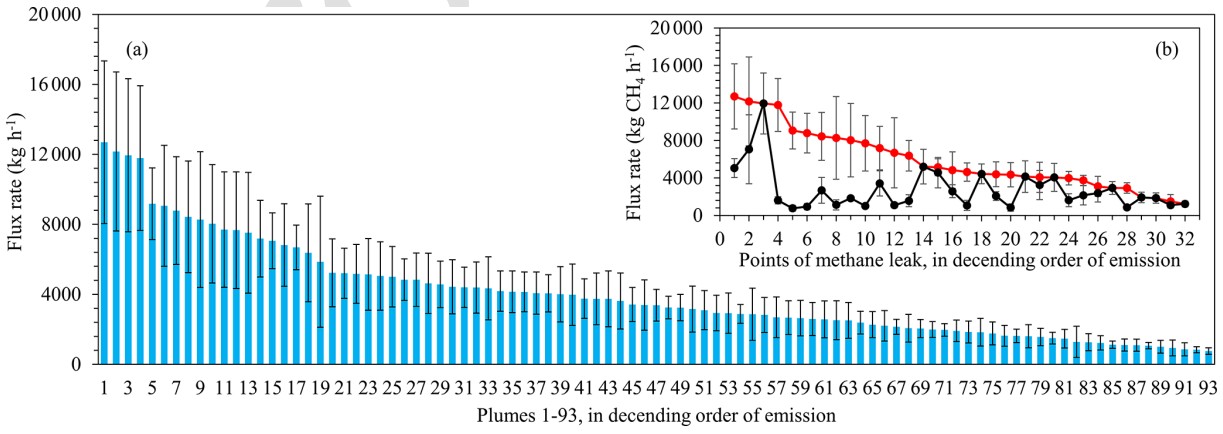

**Figure E1.** Uncertainty in the CH$_4$ emission flux rates using an absolute wind speed uncertainty. **(a)** CH$_4$ emission flux rates from point source plume nos. 1–93 in descending order of emissions, with the error bars representing the estimation uncertainty. The uncertainty in the wind speed ($1.297\,\mathrm{m\,s^{-1}}$) is estimated by comparing wind speeds from EAR5 and local meteorological stations, as described in Sect. 4.3.3. **(b)** The maximum and minimum emission flux rates for each point source with more than two observations.

*Data availability.* Gaofen-5B AHSI images are downloaded from China Centre for Resources Satellite Data and Application and accessed from https://data.cresda.cn/#/home (China Centre for Resources Satellite Data and Application, 2024). Official applications are required for accessing the GF-5B/AHSI spectra. ERA5 reanalysis data from ECMWF can be accessed from https://cds.climate.copernicus.eu/cdsapp#!/home (ECMWF, 2024). Observations from national weather station data are from China Meteorological Administration Data Centre, accessed from http://data.cma.cn/en (China Centre for Resources Satellite Data and Application, 2024). The dataset of detected plumes in Shanxi province of China during 2021–2023 using Gaofen-5B AHSI data is available at https://doi.org/10.5281/zenodo.11118337 (He et al., 2024).

*Author contributions.* ZZ designed the study. ZH prepared all datasets, carried out the retrieval, and did the result analysis. ZZ developed the retrieval codes. ZH wrote the first draft. LG and ML contributed to the data acquisition and results analysis. All authors reviewed and proofread the paper.

*Competing interests.* The contact author has declared that none of the authors has any competing interests.

ther geographical representation in this paper. While Copernicus Publications makes every effort to include appropriate place names, the final responsibility lies with the authors.

*Acknowledgements.* We thank Luis Guanter, Yao Liu, and Chenxi Sun for helpful discussions during the early stage of this work. This work has also been supported by the High-Performance Computing Platform of Peking University and China Meteorological Administration for the project of "The major Technology R&D and Application of Greenhouse Gas Observation" Youth Innovation Team (team no. CMA2023QN13) CE1.

*Financial support.* This research has been supported by the National Natural Science Foundation of Zhejiang Province (grant nos. LQ21d050001 and LZJMZ24D050008), the Research and Development Project of Zhejiang Province's "Jianbing" and "Lingyan" projects (grant no. 2023C03190), and the National Natural Science Foundation of China (grant nos. 42275142 and 12292981).

*Review statement.* This paper was edited by Jian Xu and reviewed by two anonymous referees.

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

**Remarks from the language copy-editor**

CE1     Please confirm the slight edits.