# Peer review of "A survey of methane point source emissions from coal mines in Shanxi province of China using AHSI on board Gaofen-5B"

_EGUsphere, 2023_

## Author Comment (AC1)

**A survey of methane point source emissions from coal mines in Shanxi province of China using AHSI on board Gaofen-5B**

**RC1**: 'Comment on egusphere-2023-3047', Anonymous Referee #1, 24 Jan 2024

The authors report a survey of methane plumes from underground coal mines in the Shanxi province of China using the GF-5B/AHSI hyperspectral satellite instrument. Their study builds on a growing body of literature on satellite remote sensing of methane point sources from diverse regions and industrial sectors. To the best of my knowledge, it documents the largest dataset of Shanxi coal mine methane plumes to date, including 93 plumes from 32 mines. The study is interesting and well-designed. The authors document each step of the data processing pipeline from instrument spectral calibration to characterization of wind speed errors in source rate estimation, employing the latest plume mapping/quantification techniques to uncover new emissions in a challenging study area. The study's main weakness is a lack of clarity in the description of some the methods and materials, which complicates evaluation of the conclusions. I recommend the study be accepted for publication after these issues are addressed.

We thank the reviewers for his/her constructive comments and suggestions to improve the quality and clarity of our manuscript. We have made careful modifications to the original manuscript according to all the comments and suggestions from the reviewers. The major changes include:

1. We added a paragraph to describe the new results with the wind error estimated from comparing ground-based measurements and ERA5. The comparison result shows an averaged difference of 1.297 m·s$^{-1}$, which is then used as an absolute wind uncertainty for estimating the uncertainty in emission calculation. The updated results are shown in **Section 4.3.3** and **Figure E1**.
2. We also clarified some incorrect or unclear descriptions of the results and methods throughout the manuscript. The information about the backgrounds used in the flood-fill method and the plume identification have been introduced in more details.
3. We have updated the $k$ value using the surfaced pressure that is representative for the detected plumes in Shanxi. In addition, all results in the revised paper have been updated based on this updated $k$ value.

Item-by-item responses to the specific comments are provided below, in which the reviewers' comments are in blue, our responses in **black**, and modifications of the original manuscript are indicated by highlighting in yellow in the revised manuscript.

**Comments**

L. 13-14: Please clarify what shift in center wavelength and change in FWHM are being referred to here. I believe they are departures from the nominal design values, but this could also be interpreted as temporal drift over mission operating period.

Reply: Yes, the spectral shifts and FWHM changes refer to deviations from the nominal design values. We have revised the sentence accordingly in the revised manuscript.

L. 21: It's not clear what is meant by "simultaneously". Later in the manuscript it is explained that the emissions are assumed to be simultaneous and continuous at the median observed rate. Does that include null detections? Please clarify so the reader understands from the outset.

Reply: We have removed this sentence in the revised manuscript to avoid ambiguity, since such an estimate does not include null detection and an assumption of simultaneous emission of all emitters at the detected rates may be highly uncertain. Because the temporal emission timeseries of each specific emitter is unclear, and more observations overpass at different hours would be needed for estimating the total emissions.

For example, **Chen et al. (2022a)** used high density (26292 active wells) and highly repeated (115 flight days) measurements from aerial instrument to quantify methane emissions from the whole regional study area of New Mexico Permian Basin with persistence-averaged method. The persistent emission rate from a single point source was calculated with the emission detection probability derived from highly repeated observations. In this study, this may not be feasible, because the observations are too few to calculate the possibility of emission detection. We have added the above statements in the revised manuscript.

Added reference: Chen, Y., Sherwin, E. D., Berman, E. S. F., Jones, B. B., Gordon, M. P., Wetherley, E. B., Kort, E. A., and Brandt, A. R.: Quantifying Regional Methane Emissions in the New Mexico Permian Basin with a Comprehensive Aerial Survey, Environmental Science & Technology, 56, 4317-4323, 10.1021/acs.est.1c06458, 2022a.

L. 56: "Canada's GHGSat" is a bit odd here since GHGSat is a private company whereas the other platforms in the list (belonging to Italy, China, Germany, etc.) are national/public missions.

Reply: We have revised the sentence to acknowledge that GHGSat is operated by a private company in Canada.

L. 63-65: Please specify that those 37 sources were seen in just a handful of satellite passes.

Reply: We have revised the sentence to include information about the total of 30 images acquired during several days in 2019 and 2020 (**Line 64**). This additional detail provides clarity on the extent of the observations.

L. 77: It's not clear how wind uncertainty would influence detection. I believe it's through the flood-fill algorithm but that comes later in the manuscript. Please clarify.

Reply: Thank you for pointing this out. We have removed the sentence related to the detection.

The uncertainty of wind fields does not directly affect the detection of point sources. But during the visual inspection of point sources, the wind fields will be used to assisted the identification in match the directions of plumes and winds. The uncertainty of wind fields contributes mainly

to the uncertainty in emission flux rate estimation. The flood-fill algorithm does not need the uncertainty of wind fields as input. These have been explained in the manuscript.

L. 84: Again, please clarify the nature of this shift/change.

Reply: We added the "from the nominal design" to clarify the description.

L. 97-100: Is the suggestion that the coal mine plumes in Shanxi are coming primarily from abandoned or illegal mines? That would be an interesting claim, but the satellite observations don't seem to suggest it. Have you researched the operators/practices of the 32 mines you detected? If not and to avoid this analysis I would suggest revising this passage to include other explanations (normal mine venting, for example). All coal mines are known to emit methane, including in China; I don't really see a reason to invoke abandonment or crime to explain emissions.

Reply: We appreciate your great suggestions and comments. We agree with you that such a statement is highly speculative. To address this, we have revised the description and removed the statement about abandoned or illegal mines. The revised statement is: Although the region has strict rules in regulating the process of $CH_4$, a by-product of coal mining, underground coal mines in Shanxi release $CH_4$ to the atmosphere from mine venting.

Figure 1: What is the source of the landcover imagery in panel 1a?

Reply: Thank you for your inquiry. The source of the landcover imagery in panel 1a is Google Maps, indicated by the copyright symbol (©). I have added this information to clarify the source of the background image in Figure 1.

L. 140: I don't think you defined "ILS" yet.

Reply: Instrument line shape (ILS) has been defined (ILS) earlier (**Lines 82-83**) in the manuscript.

Section 3.3.3: To clarify the IME/measurement uncertainty part of this section, please describe your implementation of the flood-fill algorithm in more detail. It wasn't clear to me, for example, that it depended on defining a background. (And what is that background? See comments below.) Can you please lay this out in a few sentences?

Reply: We added detailed information in Section 3.3.1 (**Lines 200-206**): To carry out the flood-fill method in plume extraction, a background region needs to be defined to calculate the mean and standard deviation of $\Delta XCH_4$ which set the basis for identifying anomalous high $\Delta XCH_4$ in the plume relative to the background. In this study, for a specific plume, the origin is first pinpointed through visual interpretation. Then a background region is defined as a square using the source origin as the center for calculating the mean ($\mu$) and standard deviation ($\sigma$) of $\Delta XCH_4$. Finally, a threshold defined based on $\mu$ and $\sigma$ is used for the flood-fill algorithm to effectively segment the point source plume. The exact numbers for the background square length, $\mu$ and $\sigma$ are introduced in Section 3.3.3.

And we revised the description in Section 3.3.3 (**Lines 234-237**): In practice, for estimating IME and its uncertainty for a certain plume, we used 6 different background square lengths (from 12 km to 24 km with an interval of 2.4 km) and 6 different segmentation thresholds (from $\mu+0.45\sigma$ to $\mu+0.55\sigma$ with an interval of $0.02\sigma$) for the flood-fill segmentation method (**Figure C1**). Different values of $\mu$ and $\sigma$ are calculated for different background regions.

[Figure]

**Figure C1. Examples of plume segmentation in flood-fill method using different lengths for the background square and different segmentation thresholds. Two plumes are given in $a_1$-$a_6$ and $b_1$-$b_6$ as examples, in which $a_1$-$a_3$ and $b_1$-$b_3$ are for the length of 12 km and $a_4$-$a_6$ and $b_4$-$b_6$ are for the length of 24 km. Two different thresholds, $\mu+0.45\sigma$ and $\mu+0.55\sigma$, are given for the two plume examples.**

L. 226: A flat 50% wind error would underestimate uncertainty for slow winds and overestimate uncertainty for fast winds. Why not use a fixed absolute wind uncertainty? You have local measurements to compute such an error and that would seem more defensible than an arbitrary 50%.

Reply: Thanks for your great suggestion. In the revised paper, we added a paragraph to describe the new results with the wind error estimated from comparing ground-based measurements and ERA5. The comparison result shows an averaged difference of 1.297 m·s$^{-1}$, which is then used as an absolute wind uncertainty for estimating the uncertainty in emission calculation. The updated results are shown in **Section 4.3.3** and **Figure D1**.

**Lines 410-416**: A flat 50% wind error could underestimate uncertainty for slow winds and overestimate uncertainty for fast winds. Therefore, we carried out an evaluation of the plume emission uncertainty using the absolute wind error (1.297 m·s$^{-1}$ on average) estimated by comparing wind speeds from EAR5 and local meteorological stations in Shanxi. The results of CH$_4$ flux rates and their uncertainty are shown in **Figure E1**. As we expected, the uncertainty of flux decreased/increased at high/low wind speed, respectively. In addition, the impact of wind speed uncertainties accounts for approximately 86.31%, which is similar to the previous result based on a flat 50% wind error. This result supports the fact that wind speed remains the dominant factor contributing to the uncertainty in estimating CH$_4$ point source emissions.

**Appendix E: CH$_4$ emission flux rates from point source plumes in Shanxi with an absolute wind speed uncertainty estimated by comparing wind speeds from EAR5 and local meteorological stations**

[Figure]

**Figure E1. Uncertainty of CH$_4$ emission flux rates using an absolute wind speed uncertainty. (a) CH$_4$ emission flux rates from point source plumes #1-#93 in descending order of emissions, with the error bars representing the estimation uncertainty. The uncertainty of the wind speed (1.297 m·s$^{-1}$) is estimated by comparing wind speeds from EAR5 and local meteorological stations, as described in Section 4.3.3; (b) The maximum and minimum emission flux rates for each point source with more than 2 observations.**

L. 227: "uncertainty of the used wind uncertainty". I don't understand this. Typo?

Reply: I revised it to be "the uncertainty of the used wind fields" (**Line 240**).

Figure 3: It would be helpful to note the lat/lon of the source somewhere in the caption and main text. What do the black ellipses signify? I assume the inset times are local?

Reply: The location information of the origin (lat 37°57'36", lon 113°16'04") has been added. In order to avoid any ambiguity, we removed the black ellipses and marked the original of the point sources with a red/black star. The observation time is indeed presented in standard Beijing Time (UTC+8; Lon=120°), which is close to the local time (Lon=113°). We have revised the related descriptions in the caption for **Figure 3** (**Lines 261-263**).

[Figure]

**Figure 3. Example of ΔXCH₄ retrievals from one typical point source with multiple overpasses by GF-5B/AHSI, with its origin (lat 37°57'36'', lon 113°16'04'') marked with a red/black star. The detected plumes from the seven overpasses are shown in (b)-(h). The observation times (in UTC+8 standard Beijing Time) are shown for each plume event, which are close to the local time. The background image in (a) is adopted from © Google Maps.**

L. 235: This is kind of an obvious statement – maybe not worth saying.

Reply: Thanks for pointing this out. We have removed this statement in the revised manuscript.

Section 4.1: The discussion of Figure 3 would be easier to follow if the role of the "background" in the retrieval was better explained – see above comment on Sect 3.3.3. At this point it's unclear to me whether this background is a 2D retrieval image background or a 1D spectral background. Please clarify.

Reply: **Figure 3** shows maps of ΔXCH₄ retrievals derived using **Equation 1**. It requires inputs of $\mu$ and $\Sigma$ representing the mean and covariance of the SWIR hyperspectral spectra measurements over background regions, respectively. For the ΔXCH₄ results in **Figure 3**, the spectral of the full image is used to calculate $\mu$ and $\Sigma$ for ΔXCH₄ retrieval. However, difference in the ΔXCH₄ results from two nearly simultaneously measured plumes indicate this method is not the optimal. Instead, local background spectra data would be needed in order to improve

the consistency in $\Delta XCH_4$ retrievals, as shown in **Figure 5** when overlapping background regions are used.

Therefore, for the calculation of emissions from all plumes in Shanxi, we adopted a two-step approach to identify $CH_4$ plumes and estimate emissions. In step 1, the whole image is used to calculate $\Delta XCH_4$ and identify plumes; In step 2, when implementing the flood-fill method with the strategy of selecting background regions as described in Section 3.3.3, the $\Delta XCH_4$ is re-calculated using the same background regions for the flood-fill method. In other words, the chosen background regions are used for calculating $\Delta XCH_4$ using **Equation 1**, segmenting plumes using flood-fill method, and estimating IME using **Equation 2**.

The above statements have been added in the revised manuscript in **Lines 286-290**.

It's also not clear how GF-5B/AHSI can image the same scene twice in under 10 seconds. Can you please explain the measurement strategy? Section 2.2 would be a good place for this.

Reply: The plumes are detected in the overlapping regions of two adjacent image that are observed 8 seconds apart. These two images do not cover the exact same scene. We added the following descriptions of the measurement strategy in **Section 2.2** (**Lines 112-115**): The SWIR imagery in the AHSI band employs a strategic arrangement of four strips. Each SWIR strip corresponds to a 15-km ground swath, resulting in a continuous 60 km swath width across the satellite orbit with 4 images combined. This configuration yields a total of 2,012 pixels (including 36 overlapped pixels) along the spatial dimension of the SWIR detectors (**Liu et al., 2019b**). Therefore, the target inside the overlapped pixels could be observed twice in 8 seconds.

Reference added in the revised manuscript:
Liu, Y.-N., Sun, D.-X., Hu, X.-N., Ye, X., Li, Y.-D., Liu, S.-F., Cao, K.-Q., Chai, M.-Y., Zhang, J., and Zhang, Y.: The advanced hyperspectral imager: aboard China's GaoFen-5 satellite, *IEEE Geoscience and Remote Sensing Magazine*, 7, 23-32, 2019b.

L. 255-257: The plumes sampled 8 seconds apart look the same, as expected – it's just the retrieval noise that varies between passes. Why is that? Again, please explain the observing configuration.

Reply: We explained the measurement strategy in Section 2.2 (**Lines 112-115**). The observations of the same plume sampled 8 seconds apart are different in their angles in nadir position between two strips. It leads to their different noise due to the fact that they are measured by different detectors over the edge. In addition, the background spectra used to derive $\Delta XCH_4$ are from different image scenes.

Figure 6: Please mark the source locations on the plot – in some cases I can't tell where the plume starts/ends.

Reply: We have added yellow arrows in Figure 6 to indicate the start of each point source (**Line 342 and Line 346**).

[Figure]

**Figure 6. The spatial distribution of the identified CH₄ plumes (in red dots; in total of 93) in Shanxi using GF-5B/AHSI observations, as shown in the centre panel. The black dots represent the potential point sources detected by TROPOMI (Schuit et al., 2023). CH₄ plumes (a)-(i) are examples of the identified ΔXCH₄ plumes in Shanxi and the yellow arrow points to the origins of the identified point sources. All background images ((a) – (i)) are adopted from © Google Maps.**

L. 287-292: Assuming GF-5B/AHSI observes around 11:30 local time (please specify in Section 2.2), I wonder if the difference in orbit, with TROPOMI passing ~2 hours later, is another possible explanation. It also wasn't clear to me why solar panels (which may be small compared to a TROPOMI pixel) would affect TROPOMI retrievals. Later in the manuscript, Fig. 10 shows that the solar installations are quite large, comparable in size to the TROPOMI footprint. This may be obvious to some readers, but it would be helpful to reference Fig. 10 here for others.

Reply: Thank you for your comments and suggestions. In the revised manuscript, (1) we added acknowledgement of the possibility of different overpass times between GF5B and TROPOMI (**Lines 307-309**). (2) we added descriptions related to the size of solar installations and "further details are discussed in Section 4.3.2" to refer the readers to the discussion section for more details.

L. 294-295: I don't get what point is being made here. Also, I wonder if "background spectra" is the background relevant to the flood-fill algorithm?

Reply: The background spectra is used in equation 1 to calculate ΔXCH$_4$ results, not the background in the flood-fill algorithm. The details of the background selection have been added in the revised manuscript in Section 3.3.1 and Section 3.3.3. Here, we have removed this sentence L. 294-295 to avoid ambiguity.

L. 301-302: I don't follow this. Are you saying that the IME range is bigger than the Q range, and that's because of variable wind speed?

Reply: Sorry for causing the confusion. From the flux rate distribution in Figure 7(a) and the IME distribution in Figure 7(c), we can see the order of IME does not follow that of flux rate for different point sources, indicating difference caused by the variability of wind conditions. We have rephrased this statement in the revised manuscript.

L. 304: This follows directly from the assumption of 50% wind error, so it's a foregone conclusion. I suggest estimating a typical wind error in m/s for the Shanxi area.

Reply: Thanks for your great suggestion and please see our response to your last related comment. In the revised manuscript, we have adopted a fixed absolute wind uncertainty of 1.297 m·s$^{-1}$, which is the mean difference of wind speeds from EAR5 and local meteorological stations. The detailed descriptions and results are shown in **Section 4.3.3** and **Figure E1 in the Appendix**.

L. 310-311: This argument can only be made if you include null detections in the median/mean to obtain persistence-weighted mean/median emissions for each source. Is that being done?

Reply: We have removed this sentence in the revised manuscript to avoid ambiguity, since such an estimate does not include null detection and an assumption of simultaneous emission of all emitters at the detected rates may be highly uncertain. Because the temporal emission timeseries of each specific emitter is unclear, and more observations overpass at different hours would be needed for estimating the total emissions.

For example, **Chen et al. (2022a)** used high density (26292 active wells) and highly repeated (115 flight days) measurements from aerial instrument to quantify methane emissions from the whole regional study area of New Mexico Permian Basin with persistence-averaged method. The persistent emission rate from a single point source was calculated with the emission detection probability derived from highly repeated observations. In this study, this may not be feasible, because the observations are too few to calculate the possibility of emission detection. We have added the above statements in the revised manuscript.

Added reference: Chen, Y., Sherwin, E. D., Berman, E. S. F., Jones, B. B., Gordon, M. P., Wetherley, E. B., Kort, E. A., and Brandt, A. R.: Quantifying Regional Methane Emissions in the New Mexico Permian Basin with a Comprehensive Aerial Survey, Environmental Science & Technology, 56, 4317-4323, 10.1021/acs.est.1c06458, 2022a.

Section 4.3.1: I didn't quite follow how the correction works. Does it simply substitute the estimated wavelength/FWHM values for the nominal values in the retrieval?

Reply: Yes, the estimated wavelength shift and FWHM changes relative to the nominal values are directly used as inputs in the retrieval algorithms. We emphasize this substitution in the methods of section 3.1.1.

Section 4.3.3: Earlier in the manuscript (L. 186-186), it is argued that wind direction data may be too uncertain to be useful for plume identification. But then how can it be useful for the flood-fill plume detection? This is inconsistent. Also, why not use the wind error statistics you find here instead of the flat 50% error for source rates?

Reply: The wind uncertainty has a directly impact on the uncertainty of the estimated flux rate. However, it is not directly related to plume identification and plume segmentation. Therefore, in the revised manuscript, we have removed the statements related to wind uncertainty and plume identification/segmentation.

Thanks for your great suggestion on using the wind error statistics from local meteorological stations and please see our response to your last related comment. In the revised manuscript, we have adopted a fixed absolute wind uncertainty of $1.297 \text{ m·s}^{-1}$, which is the mean difference of wind speeds from EAR5 and local meteorological stations. The detailed descriptions and results are shown in **Section 4.3.3** and **Figure E1 in the Appendix**.

**Technical corrections**

14: spectra → "spectral"

Reply: We revised it to be "spectral" in **Line 15**.

36: contribute to → "contribute"

Reply: We revised it to be "contribute" in **Line 35**.

55: EOS-1 → "EO-1"

Reply: We revised it to be "EO-1" in **Line 54**.

78: haven → "have"

Reply: We revised it to be "have" in **Line 77**.

---

## Author Comment (AC2)

RC2: 'Comment on egusphere-2023-3047', **Anonymous Referee #2, 12 Feb 2024**

He et al. used the observations of Advanced HyperSpectral Imager (AHSI) on board the Gaofen-5B satellite (GF-5B/AHSI) to estimate methane emissions from coal mines in Shanxi province in China. The spectral shift in center wavelength and change in full-width-half-maximum (FWHM) was characterized to improve the accuracy of the spectra. Based on the improved dataset, the matched filter method was applied to calculate the enhancement $\Delta XCH_4$, which is followed by the use of the integrated mass enhancement (IME) model to estimate the methane emissions. Besides these, an automated plume segmentation method was adopted to reduce the dependence of subjective judgement during the data processing phase, and the major factors that affect the uncertainties of the estimates are discussed.

We thank the reviewers for his/her constructive comments and suggestions to improve the quality and clarity of our manuscript. We have made careful modifications to the original manuscript according to all the comments and suggestions from the reviewers. The major changes include:

1. We added a paragraph to describe the new results with the wind error estimated from comparing ground-based measurements and ERA5. The comparison result shows an averaged difference of 1.297 m·s$^{-1}$, which is then used as an absolute wind uncertainty for estimating the uncertainty in emission calculation. The updated results are shown in **Section 4.3.3** and **Figure E1**.
2. We also clarified some incorrect or unclear descriptions of the results and methods throughout the manuscript. The information about the backgrounds used in the flood-fill method and the plume identification have been introduced in more details.
3. We have updated the $k$ value using the surfaced pressure that is representative for the detected plumes in Shanxi. In addition, all results in the revised paper have been updated based on this updated $k$ value.

Item-by-item responses to the specific comments are provided below, in which the reviewers' comments are in blue, our responses in **black**, and modifications of the original manuscript are indicated by highlighting in yellow in the revised manuscript.

**General comments:**

1. It is apparent that the estimates are sensitive to the selection of background, which is acknowledged in the manuscript, and is also demonstrated with a good example using the same background region in Fig.5a&b. The question is how the background regions were selected in practice? i.e., where is "a background region in square (with length of 600 pixels, which is 18 km)" located?

Reply: We added the detailed information in Section 3.1.1 (**Lines 200-206**): To carry out the flood-fill method in plume extraction, a background region needs to be defined to calculate the mean and standard deviation of $\Delta XCH_4$ which set the basis for identifying anomalous high

$\Delta XCH_4$ in the plume relative to the background. In this study, for a specific plume, the origin is first pinpointed through visual interpretation. Then a background region is defined as a square using the source origin as the center for calculating the mean ($\mu$) and standard deviation ($\sigma$) of $\Delta XCH_4$. Finally, a threshold defined based on $\mu$ and $\sigma$ is used for the flood-fill algorithm to effectively segment the point source plume. The exact numbers for the background square length, $\mu$ and $\sigma$ are introduced in Section 3.3.3.

We have also revised the related descriptions in **Lines 234-237**: In practice, for estimating IME and its uncertainty for a certain plume, we used 6 different background square lengths (from 12 km to 24 km with an interval of 2.4 km) and 6 different segmentation thresholds (from $\mu+0.45\sigma$ to $\mu+0.55\sigma$ with an interval of 0.02 $\sigma$ ) for the flood-fill segmentation method (**Figure C1**). Different values of $\mu$ and $\sigma$ are calculated for different background regions.

We have also added **Figure C1** in the Appendix to demonstrate examples of background selection.

[Figure]

**Figure C1. Examples of plume segmentation in flood-fill method using different lengths for the background square and different segmentation thresholds. Two plumes are given in $a_1$-$a_6$ and $b_1$-$b_6$ as examples, in which $a_1$-$a_3$ and $b_1$-$b_3$ are for the length of 12 km and $a_4$-$a_6$ and $b_4$-$b_6$ are for the length of 24 km. Two different thresholds, $\mu+0.45\sigma$ and $\mu+0.55\sigma$, are given for the two plume examples.**

2. Are the estimates of methane emissions correlated with wind speed? As the wind speed is low, the dispersion of the methane plumes may be very uncertain, and the estimates may be biased.

Reply: Based on the wind speeds from ERA5 and the corresponding $CH_4$ emission estimations, we made a scatter plot as shown in the following **Figure**. No clear correlation can be seen from

the data, suggesting the emission estimations are not biased due to wind speeds. We have added the following figure and related statements in the revised manuscript.

[Figure]

**Figure. Scatter plot between the estimates of methane emissions and wind speed in this study.**

3. I understand that the overall emission flux rate of 13.26 ton h-1 in Shanxi refers to the 32 coal mines between 2021 and 2023. How does it relate to the total coal mine emissions? How does it compare with the estimates from TROPOMI? What's the detection limit of GF-5B/AHSI?

Reply: We have removed this sentence in the revised manuscript to avoid ambiguity, since such an estimate does not include null detection and an assumption of simultaneous emission of all emitters at the detected rates may be highly uncertain. Because the temporal emission timeseries of each specific emitter is unclear, and more observations overpass at different hours would be needed for estimating the total emissions.

For example, **Chen et al. (2022a)** used high density (26292 active wells) and highly repeated (115 flight days) measurements from aerial instrument to quantify methane emissions from the whole regional study area of New Mexico Permian Basin with persistence-averaged method. The persistent emission rate from a single point source was calculated with the emission detection probability derived from highly repeated observations. In this study, this may not be feasible, because the observations are too few to calculate the possibility of emission detection. We have added the above statements in the revised manuscript.

We have made a comparison of emissions estimated from TROPOMI (**Schuit et al., 2023**) and this study as shown in the following **Figure**. Notably, the emission rate of plumes from GF-5B AHSI are very different from results by TROPOMI. For the detection limit, the emission rates of all detected plume are shown in this study (Figure 7), we can see that the smallest plume has an emission rate of 761.78 ± 185.00 kg·h$^{-1}$ which may represent the detection limit of GF-5B AHSI.

[Figure]

[Figure]

**Figure. Comparison of point source emission statistics between TROPOMI (a) and GF-5B AHSI (b) in Shanxi, China. TROPOMI detected a total of 134 points (Schuit et al., 2023), while GF-5B AHSI identified 93 plumes in this study.**

Added reference: Chen, Y., Sherwin, E. D., Berman, E. S. F., Jones, B. B., Gordon, M. P., Wetherley, E. B., Kort, E. A., and Brandt, A. R.: Quantifying Regional Methane Emissions in the New Mexico Permian Basin with a Comprehensive Aerial Survey, Environmental Science & Technology, 56, 4317-4323, 10.1021/acs.est.1c06458, 2022a.

**detailed comments:**

L138-139: the matched filter method and the IME model are actually combined to estimate the $CH_4$ emission rates. Therefore, they should not be considered two methods.

Reply: Thank you for your comment. You are correct that in our study, the matched filter method and the IME model are indeed combined to estimate $CH_4$ emission rates. We have revised the text accordingly to accurately reflect this aspect of our methodology in **Lines 139-140**.

L184: change "be very differ" to "be very different", what are the possible reasons of the mismatch?

Reply: We changed "be very differ" to be "be very different" in **Line 186**. The reasons for the mismatch are explained in **Lines 187-188**: Several factors could contribute to this mismatch, including the temporal and spatial resolution of the reanalysis data, local topographical features, and microscale meteorological phenomena that are not fully captured by the reanalysis data.

L204-205: k may be affected by a few factors, such as surface pressure, temperature, and water vapor. How would these affect the estimate?

Reply: Thank you for your great suggestion, we have updated the $k$ value using the surfaced pressure that is representative for the detected plumes in Shanxi. In addition, all results in the revised paper have been updated based on this updated $k$ value.

We revised the related descriptions in **Lines 215-221**: $k$ is the scaling factor converting $\Delta XCH_4$ from volume mixing ratio to mass based on Avogadro's law, considering the pixel resolution of GF-5B/AHSI to be 30-meter. In **Guanter et al. (2021)**, k is defined as $5.155 \times 10^-$

[3] kg·ppb-1derived from surface pressure of one standard atmosphere. However, the average elevation of the identified plumes is 942.41 meters (**Figure B1**), whose surface pressure (900.64 hPa) is about 10% less than one standard atmosphere. Therefore, we calculated a new k based on the derived averaged surface pressure for all the identified plumes. The derived k value ($4.565 \times 10^{-3}$ kg·ppb$^{-1}$) is then used for estimating IME in this study.

All the IME and $CH_4$ emission rates throughout the revised manuscript were updated with the new scaling factor of $k$ used in this study.

[Figure]

**Figure B1. Histogram of the elevation for the detected plumes in Shanxi. The elevation data is from the DEM shown in Section 2.3.**

L227: change "evaluations" to "evaluation"

Reply: We changed the "evaluations" to be "evaluation" in **Line 241**.

L235: not "emissions" but "the direction of plumes"

Reply: We have revised the statement to emphasize the significance of the observed differences and the importance of repeated observations for accurate emission estimation (**Lines 249-250**).

L241-243: "as the plumes appears at different locations of the imaging scene. The plumes appear at the bottom of the scene in Figure (f) and at the top in Figure (g)", why does the position of the plumes in the scenes matter?

Reply: The retrieval of $\Delta XCH_4$ relies on the quality of the spectra, as quantified by its signal to noise ratio (SNR). When plumes appear at different locations of the imaging scene (as illustrated in the following **Figure D1**), they may be observed by different detectors with different SNR of the instrument. Therefore, as we explained in the manuscript, the difference in $\Delta XCH_4$ may be slightly caused by the different signal noise ratio of the detectors.

We have changed the statement in the revised manuscript to:
"as the plumes appear at different locations of the imaging scene. They may be observed by different detectors with different SNR of the instrument that affect the detection accuracy of the plumes."

[Figure]

**Figure D1. The full images of ΔXCH4 for Figure 3f (a) and Figure 3g (b). The plume target (pointed by the white arrow) appears at the overlapping region of the two images.**

L310:   13.26 t*24*365 /yr

Reply: In the revised manuscript, we remove summation of the total emissions in Shanxi based on the assumption of simultaneous emission to avoid ambiguity, since such an estimate does not include null detection and an assumption of simultaneous emission of all emitters at the detected rates may be highly uncertain. Because the temporal emission timeseries of each specific emitter is unclear, and more observations overpass at different hours would be needed for estimating the total emissions. Related statements have been added to the revised manuscript.

L356: consider combining or to combine

Reply: We changed the "combine" to be " combining" in **Line 384**.